# Sulfated glycosaminoglycans inhibit LCMV entry and modulate antiviral immunity and pathology

Michal Gorzkiewicz [1,2,11], Soha Noseir[1,11], Mandar Vengurlekar[1], Mitrajit Ghosh[1], Ichiro Katahira [1], Džiuljeta Abromavičiūtė [1], Ulla Gerling-Driessen [3], Lorand Bonda[4], Nick Rähse [5], Marco Lapsien[5], Sabrina Bockholt[6], Ann Kathrin Bergmann [7], Konstantina Kostadinovska[1], Hafssa Fraii [1], Karl S Lang[8], Lisa Oestereich[6,9], Holger Gohlke[5,10], Laura Hartmann[3] & Philipp A Lang [1]✉

## Abstract

Viral infections remain a major challenge due to the limited availability and efficacy of current treatments. Existing antivirals primarily target viral replication but are often virus-specific and can lead to drug resistance. Sulfated glycosaminoglycans (GAGs) have emerged as promising broad-spectrum agents that block viral binding and entry into host cells. Here, we show that highly sulfated GAGs restrict the infectivity of both pathogenic and non-pathogenic Arenaviruses. Using the lymphocytic choriomeningitis virus (LCMV) model, we demonstrate that GAG exposure reduces viral entry and infection in cell lines and bone marrow-derived dendritic cells, impairing their ability to activate antiviral T cells. In vivo, early exposure of LCMV to dextran sulfate suppressed immune activation, leading to diminished T-cell responses, prolonged infection, and increased immunopathology. By contrast, administering dextran sulfate during the acute infection phase decreased viral load, improved effector T-cell function, and reduced liver pathology. These findings highlight the therapeutic potential of sulfated GAGs against Arenavirus infections and the importance of treatment timing for clinical efficacy.

Keywords GAGs; LCMV; Arenavirus; Dextran Sulfate; Infection
Subject Categories Microbiology, Virology & Host Pathogen Interaction; Pharmacology & Drug Discovery

## Introduction

Despite significant efforts and expanding biomedical knowledge, infectious diseases remain a significant global health burden (Baker et al, 2022). Arenaviruses are enveloped, bi-segmented single-stranded RNA viruses, with numerous identified species. These pathogens are associated with rodent-transmitted diseases in humans, from asymptomatic and mild infections to life-threatening conditions, including South American hemorrhagic fevers and Lassa fever (McLay et al, 2014). Because of their epidemic potential and high case-fatality rates, species such as the Lassa virus and the Junin virus are intensively studied to establish vaccine and therapy regimens (Hastie et al, 2023). At the same time, treatment options for Arenaviruses remain limited.

Lymphocytic choriomeningitis virus (LCMV) serves as a well-established model for studying arenaviral biology and immune activation. While LCMV infections in humans are typically asymptomatic, severe outcomes have been documented in immunosuppressed patients in the context of organ transplantation or following vertical transmission during pregnancy (Bonthius et al, 2007; Amman et al, 2007; Emonet et al, 2007). LCMV variants can be classified as low or high-affinity variants based on the mutations in their glycoprotein (GP) sequence that modulate binding to LCMV's primary cellular receptor, α-dystroglycan (α-DG) (Smelt et al, 2001; Hastie et al, 2016; Gorzkiewicz et al, 2023). LCMV variants have been pivotal in elucidating various aspects of T-cell biology, including activation, exhaustion, and memory development (Kahan and Zajac 2019).

Like many other viruses, Arenaviruses can engage cell-surface proteoglycans during the initial attachment step (García et al, 2016; De Pasquale et al, 2021; Volland et al, 2021). Proteoglycans are present on the surface of almost all cells, consisting of core proteins that anchor them in the cellular membrane, and heterogeneous glycosaminoglycan (GAG) side chains (Iozzo and Schaefer, 2015;

[1]Department of Molecular Medicine II, Medical Faculty and University Hospital, Heinrich Heine University Düsseldorf, Universitätsstr. 1, Düsseldorf 40225, Germany. [2]Department of General Biophysics, Faculty of Biology and Environmental Protection, University of Lodz, 141/143 Pomorska St., Lodz 90-236, Poland. [3]Institute for Macromolecular Chemistry, University of Freiburg, Stefan-Meier-Str. 31, Freiburg 79104, Germany. [4]Department of Organic and Macromolecular Chemistry, Heinrich Heine University Düsseldorf, Universitätsstr. 1, Düsseldorf 40225, Germany. [5]Institute for Pharmaceutical and Medicinal Chemistry, Heinrich Heine University Düsseldorf, Universitätsstr. 1, Düsseldorf 40225, Germany. [6]Bernhard Nocht Institute for Tropical Medicine, Bernhard-Nocht-Str. 74, Hamburg 20359, Germany. [7]Core Facility Electron Microscopy, Medical Faculty and University Hospital, Heinrich Heine University, Universitätsstr. 1, Düsseldorf 40225, Germany. [8]Institute of Immunology, University Hospital Essen, Hufelandstr. 55, Essen 45122, Germany. [9]German Center for Infection Research, Partner Site Hamburg-Lübeck-Borstel-Riems, Hamburg, Germany. [10]Institute of Bio- and Geosciences (IBG-4: Bioinformatics), Forschungszentrum Jülich GmbH, Wilhelm-Johnen-Straße, Jülich 52428, Germany. [11]These authors contributed equally: Michal Gorzkiewicz, Soha Noseir. ✉E-mail: langp@uni-duesseldorf.de

Pomin and Mulloy, 2018). Their negatively charged sulfated residues mediate largely electrostatic interactions with diverse proteins (Kjellén and Lindahl, 2018; Crijns et al, 2021). The crucial role of GAGs in facilitating virus-host cell interactions prompted studies on using their soluble forms as "decoy receptors". Sulfated glycosaminoglycans, including heparan sulfate and its analogs, are of particular interest because their unique biochemical properties, particularly their dense negative charge, enable efficient interactions with viral proteins, thus preventing their binding to specific cell surface receptors (Rusnati and Lembo, 2016). The antiviral effect of sulfated polysaccharides and other polyanions has already been documented for different virus species (Bello-Morales et al, 2022), such as HSV (Gangji et al, 2018) and HIV (Connell and Lortat-Jacob, 2013). Early studies also suggested that sulfated polysaccharides can inhibit Arenavirus replication (Andrei and De Clercq, 1990), but the mechanisms and extent of this inhibition, as well as its in vivo relevance, remain incompletely understood. Given their ubiquity on host cells, structural diversity, and the feasibility of generating sulfated polysaccharide derivatives with defined characteristics, sulfated GAGs represent an especially relevant candidate class of molecules to study in the context of Arenavirus entry and potential therapeutic intervention. Considering these aspects, we hypothesized that sulfated glycans with varying degrees of sulfation might influence Arenavirus-host interactions and cell entry, contributing to the outcome of infection in vivo. We show that highly sulfated GAGs block LCMV cell entry in vitro, reducing viral replication and titers, and that this effect is consistent across different pathogenic and non-pathogenic Arenaviruses. Dextran sulfate, identified as the most potent inhibitor, reduced the viral load in spleen and liver tissue and decreased liver pathology when administered during the acute phase of LCMV infection in mice. This effect was associated with an increased number of LCMV-specific effector T cells, reduced T-cell exhaustion, and enhanced cytokine production. Taken together, these findings highlight the potential of sulfated GAGs as therapeutic agents for Arenavirus infections and open up avenues for novel sugar-based antiviral strategies.

# Results

## Heparin inhibits LCMV cellular entry and replication in vitro

In order to investigate the influence of heparin treatment during LCMV WE infection, MC57G cells were pretreated with the compound and infected, with subsequent time-dependent Golgi blocking, followed by intracellular LCMV NP staining. Heparin inhibited the frequency of LCMV NP$^+$ cells in a concentration-dependent manner at all time points tested (Fig. 1A–C). Consistently, decreased expression levels of viral RNA were determined in heparin-treated cells, when compared to controls (Fig. 1D,E; Appendix Fig. S1A,B). When LCMV NP and GP were stained in those samples, a reduced frequency of LCMV-infected cells was observed following heparin treatment (Fig. 1F; Appendix Fig. S1C). Consequently, decreased viral titers in culture supernatants after heparin treatment were detected when compared to supernatants of control cells (Fig. 1G). Next, we wondered whether pretreatment of virus particles would give similar results, namely reduced viral

entry following LCMV infection. The effect of virus pretreatment on cell entry was consistent with the observations obtained during treatment of cells (Fig. 1H–J). Hence, the latter experimental setup was continued in further in vitro experiments.

## Inhibition of arenaviral infection by GAGs depends on their structure and sulfation

To further investigate the antiviral potential of differentially sulfated polysaccharides, an in vitro screening using dextran, ulvan, chondroitin sulfate A (chondroitin-4-sulfate), chondroitin sulfate SC (a mixture of chondroitin-4-sulfate and chondroitin-6-sulfate), dextran sulfate (5 kDa and 9–20 kDa), and hyaluronic acid (15 kDa and 50 kDa) was performed (Fig. EV1A). The sulfur content in studied compound samples was determined by elemental analysis (Fig. 2A). As expected, the treatment of MC57G cells prior to infection resulted in differential inhibition of LCMV replication, suggesting a dependence on the level of sulfation of tested GAGs (Fig. 2B). Specifically, non-sulfated compounds (dextran and hyaluronic acids) showed no-to-limited inhibition at highest concentration tested (5000 μg/ml), and dextran sulfates exhibited the highest inhibitory activity, with the effect being observed in the entire concentration range. Interestingly, ulvan showed a more potent antiviral effect than chondroitin sulfates, despite similar sulfur content (Fig. 2B). This could be attributed to slight differences in the chemical structure of these compounds: compared to ulvan, chondroitins contain an additional NHCOCH$_3$ moiety, which could hamper the binding to the virus and/or cell surface. Nevertheless, in the course of further experiments, we focused on dextran sulfate (9–20 kDa) as the most promising antiviral agent, with non-sulfated dextran serving as a negative control (Fig. 2B). As expected, dextran sulfate was able to significantly decrease the level of viral transcripts in infected cells compared to non-sulfated dextran (Fig. EV1B–E).

In order to confirm that observed antiviral activity is indeed based on sulfation status, we performed in vitro chemical reactions for global sulfation of dextran used in the control setting (Fig. 2C). When we pre-incubated BHK cells with non-sulfated dextran, we found no inhibition of LCMV entry (Fig. 2D). In sharp contrast, commercially available dextran sulfate (Fig. 2E) and dextran sulfated in-house (Fig. 2F) could prevent LCMV entry into BHK cells. Both dextran sulfates exhibited concentration-dependent inhibition to a similar degree. Here, it should be noted that observed effects did not result from the cytotoxicity of studied compounds, as both MC57G and BHK cells showed high levels of viability after GAG treatment in the whole concentration range (Appendix Fig. S2).

Furthermore, we synthesized a mannose-based glycopolymer and its sulfated analog (Fig. 2G). Similar to the analyses with dextran/dextran sulfate, the sulfated polymer efficiently inhibited the entry of LCMV WE into BHK cells in a concentration- and time-dependent manner (Figs. 2H and EV1F). These data indicate that sulfation of glycosaminoglycans might be critical for exhibiting anti-arenaviral effects.

Next, we extended our research to include other Arenaviruses in order to investigate whether the observed antiviral effect of sulfated GAGs is virus-specific. MC57G cells were treated with heparin, dextran, or dextran sulfate, and infected with Lassa (LASV), Lujo (LUJV), Junin (JUNV), Machupo (MACV), Tacaribe (TCRV),

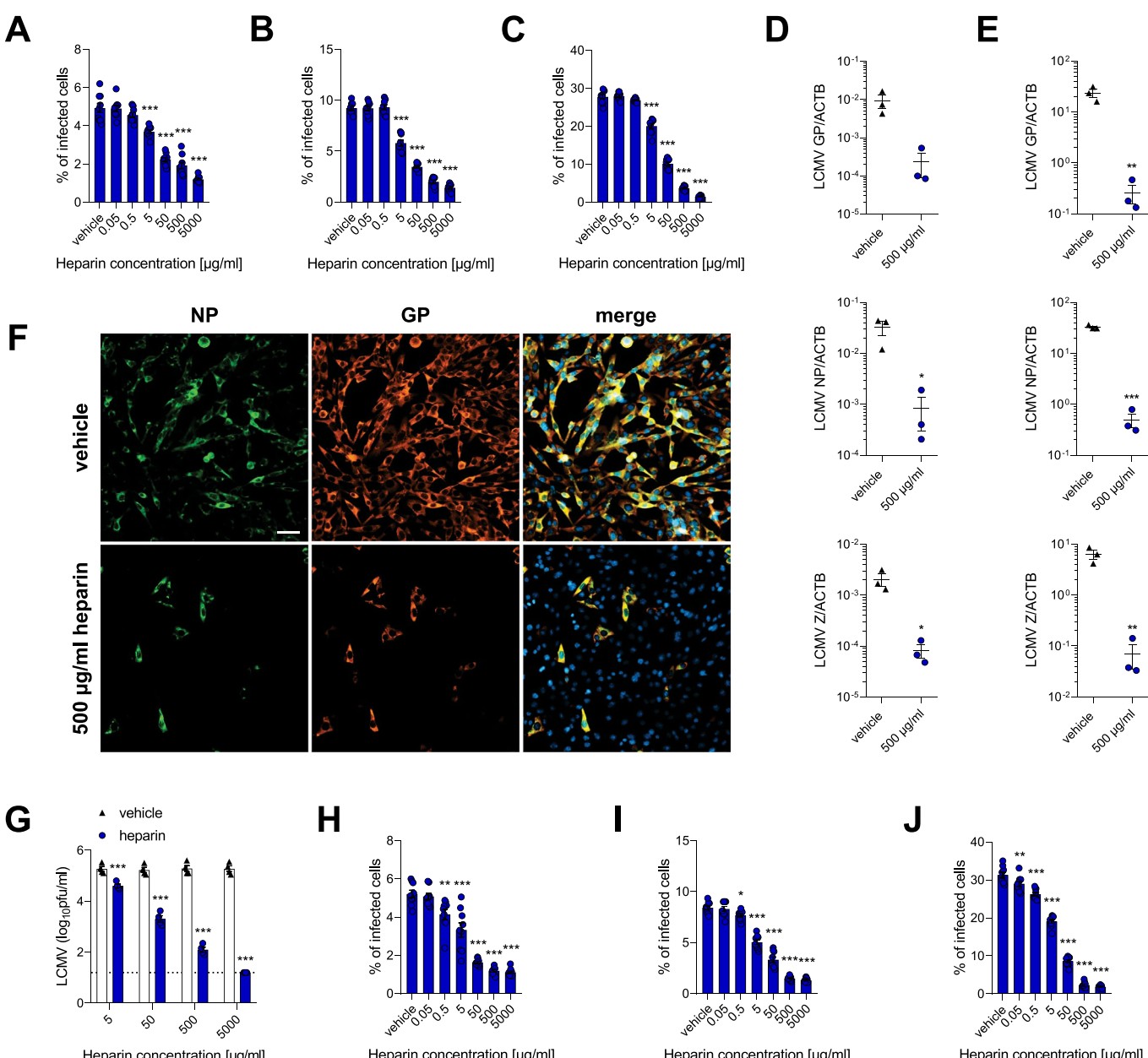

**Figure 1. Heparin decreases LCMV cell entry and replication.**

(**A–C**) MC57G cells were treated with heparin for 1 h. Next, cells were incubated with LCMV WE (MOI 0.5) for 1 h at 4 °C. To initiate infection, cells were moved to 37 °C for 2.5 (**A**), 5 (**B**), and 24 h (**C**) before adding monensin to block the Golgi apparatus and intracellular protein transport. Cells were then stained for viability and LCMV NP. Data presented as mean ± SEM, n = 8–9. ***P < 0.001 compared to vehicle control. Statistical significance was assessed by one-way ANOVA. (**D, E**) MC57G cells were treated with heparin for 1 h. Next, cells were infected with LCMV WE (MOI 0.1) for 3 h. After the infection, cells were washed three times with PBS, and fresh medium containing heparin was re-added to the cells. The cells were collected after 8 (**D**) and 24 h (**E**) for RNA isolation and RT-PCR. Data presented as number of cognate mRNA copies per copy of mRNA for reference housekeeping gene, mean ± SEM, n = 3. *P < 0.05, **P < 0.01, ***P < 0.001 compared to vehicle control. Statistical significance was assessed by Student's t test. (**F, G**) MC57G cells were treated with heparin for 1 h. Next, cells were infected with LCMV WE (MOI 0.1) for 3 h. After the infection, cells were washed three times with PBS and fresh medium containing heparin was re-added to the cells. After 24 h, the cells were stained for LCMV NP and GP (**F**, one representative set of n = 4 is shown; scale bar = 50 μm). Supernatant from infected cells was used to determine viral titer via plaque assay (**G**). Data presented as mean ± SEM, n = 4. ***P < 0.001 compared to vehicle control. Statistical significance was assessed by two-way ANOVA. (**H–J**) LCMV WE was treated with heparin for 1 h. Next, MC57G cells were incubated with heparin-pretreated virus (MOI 0.5) for 1 h at 4 °C. To initiate infection, cells were moved to 37 °C for 2.5 (**H**), 5 (**I**), and 24 h (**J**) before adding monensin to block the Golgi apparatus and intracellular protein transport. Cells were then stained for viability and LCMV NP according to the standard protocol. Data presented as mean ± SEM, n = 9. *P < 0.05, **P < 0.01, ***P < 0.001 compared to vehicle control. Statistical significance was assessed by one-way ANOVA. Source data are available online for this figure.

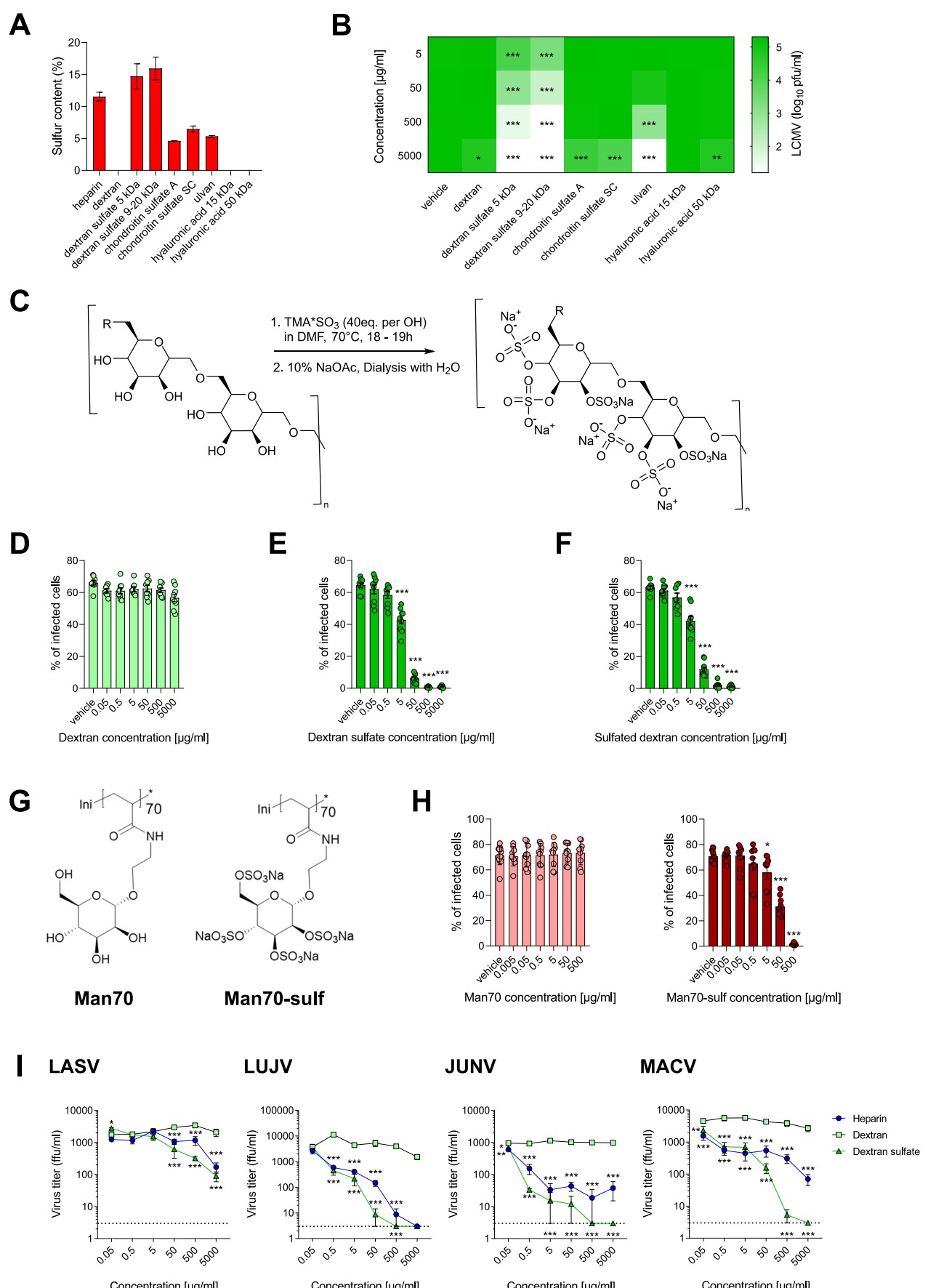

**Figure 2.   Inhibition of Arenavirus infection by GAGs depends on their structure and sulfation.**

(**A**) Sulfur content in the studied GAG samples, presented as percentage of total elemental composition (mean ± SEM, $n = 2$). (**B**) MC57G cells were treated with GAGs for 1 h. Next, cells were infected with LCMV WE (MOI 0.1) for 3 h. After the infection, cells were washed three times with PBS, and fresh medium containing GAGs was re-added to the cells. The supernatant was collected after 24 h from infected cells to perform a plaque assay. Data presented as heatmap representing mean, $n = 4$. *$P < 0.05$, **$P < 0.01$, ***$P < 0.001$ compared to vehicle control. Statistical significance was assessed by two-way ANOVA. (**C**) Reaction scheme and conditions for global sulfation of dextran. (**D-F**) BHK cells were treated for 1 h with dextran (**D**), commercial dextran sulfate (**E**), or dextran chemically sulfated in-house (**F**). Next, cells were infected with LCMV WE (MOI 0.5) for 1 h at 4 °C. After the infection, cells were incubated at 37 °C for 24 h before adding monensin to block the Golgi apparatus and intracellular protein transport. Cells were then stained for viability and LCMV NP according to the standard protocol. Data presented as mean ± SEM, $n = 7$–9. ***$P < 0.001$ compared to vehicle control. Statistical significance was assessed by one-way ANOVA. (**G**) Chemical structures of mannose polymers in non-sulfated and sulfated form. (**H**) BHK cells were treated with Man70 or Man70-sulf for 1 h. Next, cells were infected with LCMV WE (MOI 0.5) for 1 h at 4 °C. After the infection, cells were incubated at 37 °C overnight before adding monensin to block the Golgi apparatus and intracellular protein transport. Cells were then stained for viability and LCMV NP according to the standard protocol. Data presented as mean ± SEM, $n = 9$–12. *$P < 0.05$, ***$P < 0.001$ compared to vehicle control. Statistical significance was assessed by one-way ANOVA. (**I**) MC57G cells were treated with heparin, dextran, or dextran sulfate, and infected with Lassa (LASV), Lujo (LUJV), Junin (JUNV), and Machupo (MACV) viruses (BSL4) at MOI of 0.001. Cell culture supernatant was harvested after 2 days, and virus titers were determined by immunofocus assay. Data presented as focus-forming units (ffu) per ml, mean ± SEM $n = 3$. *$P < 0.05$, **$P < 0.01$, ***$P < 0.001$ compared to dextran treatment. Statistical significance was assessed by two-way ANOVA. Source data are available online for this figure.

Parana (PARV), and Morogoro (MORV) viruses. In line with previous results, dextran did not inhibit the virus titer in supernatants, while both heparin and dextran sulfate showed antiviral effect against both pathogenic (Fig. 2I) and non-pathogenic (Fig. EV1G) Old and New World Arenaviruses. Different viruses exhibited varying sensitivity towards GAGs, with the strongest inhibitory effect observed for LUJV and JUNV. Taken together, these data point to potent antiviral effects of sulfated GAGs, particularly dextran sulfate, against several Arenavirus strains.

To evaluate this phenomenon more closely, we analyzed early LCMV WE entry into MC57G cells pretreated with dextran and dextran sulfate. In this experimental setup, MC57G cells were infected with LCMV WE and the viral cell entry was stopped after 1, 5, and 15 min by placing the cells on ice, with subsequent processing of the samples for analyses by transmission electron microscopy (TEM). Such an approach enabled visualization of viral particles in the vicinity of the cells and in the intercellular spaces, both after dextran (Fig. 3A) and dextran sulfate treatment (Fig. 3B). With increasing time of infection, a decrease in the number of viral particles outside the cell was observed in dextran-treated cells, an effect that was not determined in case of dextran sulfate treatment (Fig. 3C). This outcome was confirmed by the determination of viral transcripts in an analogous experimental setup (Fig. 3D). Thus, we hypothesized that sulfated glycopolymers may inhibit arenaviral infection in vitro by limiting its cell binding and subsequent cellular entry. To investigate the interaction further, we pretreated LCMV WE with dextran sulfate and removed excess unbound GAG by filtration through 100 kDa filters. Subsequent entry assay in MC57G cells showed that filtration itself does not directly influence the infectivity of LCMV, while the majority of dextran sulfate remains bound to the virus, providing the inhibition of infection (Fig. 3E). Moreover, we pretreated MC57G cells with dextran and dextran sulfate, and subsequently treated them with recombinant LCMV GP1-Fc. Following washing steps and additional staining with PE-conjugated anti-Fc antibody, we visualized (Fig. 3F) and quantified (Fig. 3G) cell-bound LCMV GP. Also in this case, dextran sulfate showed potent inhibition of LCMV GP1-Fc binding. In conclusion, these data suggest that dextran sulfate interacts with the virus, thus preventing its binding to the cell surface and subsequent infection.

## Dextran sulfate can interact with LCMV GP

Since Tyr155 is associated with high-affinity binding to α-DG (Hastie et al, 2016; Volland et al, 2021; Xu et al, 2024), we used two recombinant LCMV variants, carrying His155 (rWT), and Tyr155 (H155Y variant) in the GP sequence, in order to evaluate the inhibitory potential of dextran sulfate towards them (Fig. 4A,B; Appendix Fig. S3A–D). We observed that in both cases, dextran sulfate was able to inhibit viral cell entry, but the kinetics of inhibition and the range of effective concentrations slightly differed: dextran sulfate showed higher efficacy against the virus with lower receptor affinity, indicated by lower IC50 concentrations. In addition, we performed an entry assay of LCMV WE and clone 13 (the latter with high α-DG affinity) in WT and α-DG-deficient HEK293T cells, and in Vero cells, which lack glycosylated α-DG (Shimojima et al, 2012). These data indicated that regardless of cell type and virus entry rate, dextran sulfate was still able to inhibit the viral cell entry in all cases (Appendix Fig. S3E–J). Once again, dextran sulfate was more efficient toward the low-affinity virus (LCMV WE), in line with the effect observed for rWT and H155Y variants.

In order to elucidate a possible mode of action for the observed experimental results, an atomistic model of the fully glycosylated (Appendix Fig. S4) pre-fusion LCMV GP trimer (PDB ID: 8DMI) (Moon-Walker et al, 2023) was generated and subjected to unbiased all-atom free ligand diffusion molecular dynamics (fldMD) simulations. We performed 10 independent simulations for the sulfated dextran (5 kDa) and 10 for the non-sulfated dextran (negative control), each 1.5 µs long. The dextran molecules were initially placed at least 40 Å from the GP surface at a random positions within the simulation box. All simulations were conducted without prior knowledge of the binding site location.

In the majority of the simulations, the ligands diffused extensively before interacting with the GP. Once bound, the ligands remained attached for the remainder of the simulation (Appendix Fig. S5). Since we used the pre-fusion state of the LCMV GP trimer for the simulations, all interactions with the C-terminal domain of GP2 were neglected. This domain would normally be embedded in the membrane and, therefore, be inaccessible to the ligand. In all 10 simulations, the non-sulfated dextran polymer

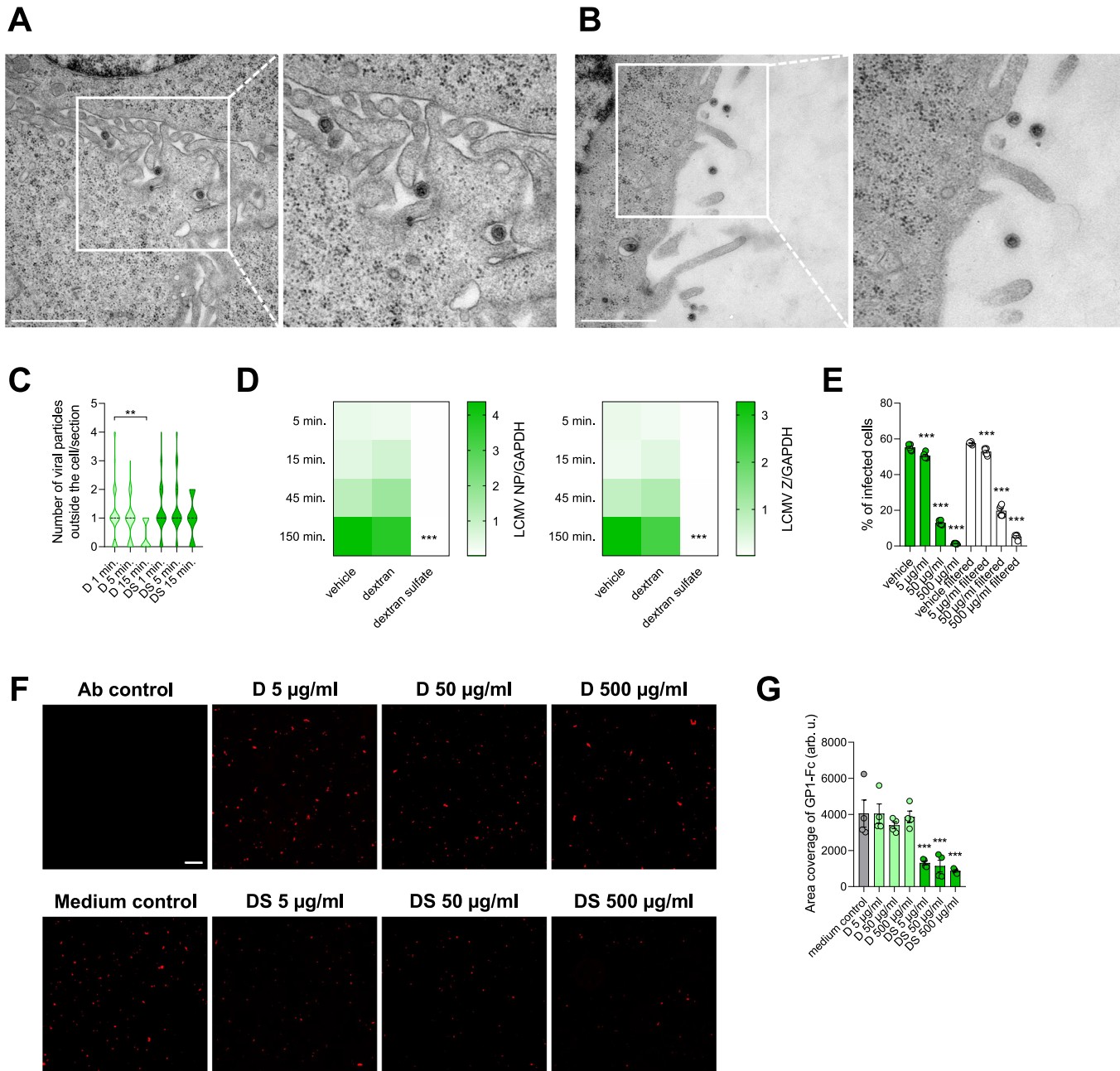

**Figure 3. Dextran sulfate influences viral cell entry at early stages of infection.**

(A, B) MC57G cells were treated with dextran and dextran sulfate (500 µg/ml) for 1 h and subsequently infected with LCMV WE (MOI 10) for 1, 5, and 15 min. After the indicated time points, cells were collected, fixed, and processed for TEM analysis. Pictures of viral particles in the vicinity of cells treated with dextran (A) and dextran sulfate (B) after 1 min of infection are shown. Scale bar = 1 µm. (C) Violin plot showing the distribution of the number of viral particles outside the cell per section, $n = 36$–46 from two independent experiments. **$P < 0.01$. Statistical significance was assessed by one-way ANOVA. (D) MC57G cells were treated with dextran and dextran sulfate (500 µg/ml) for 1 h. Next, cells were infected with LCMV WE (MOI 0.1) for 5, 15, 45, and 150 min. After the infection, cells were washed three times with cold PBS and lysed with TRIzol for RNA isolation and RT-PCR. Data presented as a heatmap representing the mean number of cognate mRNA copies per copy of mRNA for the reference housekeeping gene, $n = 4$. ***$P < 0.001$ compared to vehicle control. Statistical significance was assessed by two-way ANOVA. (E) LCMV WE was treated with dextran sulfate for 1 h. Next, the samples were filtered through Amicon® Ultra-0.5 Centrifugal Filter Device (100 kDa) three times to remove unbound dextran sulfate. MC57G cells were subsequently incubated with filtered and non-filtered virus samples (MOI 0.5) for 1 h at 4 °C. To initiate infection, cells were moved to 37 °C for 24 h before adding monensin to block the Golgi apparatus and intracellular protein transport. Cells were then stained for viability and LCMV NP according to the standard protocol. Data presented as mean ± SEM, $n = 6$. ***$P < 0.001$ compared to respective vehicle control. Statistical significance was assessed by one-way ANOVA. (F, G) MC57G cells were pretreated with dextran and dextran sulfate, and subsequently treated with supernatant derived from HEK293 cells expressing LCMV GP1-Fc. Following staining with PE-conjugated anti-Fc antibody (Ab), cell-bound LCMV GP was visualized under a fluorescence microscope. Representative images (F, scale bar = 50 µm) and quantification (G) are presented (mean ± SEM, $n = 4$. ***$P < 0.001$ compared to medium control). Statistical significance was assessed by one-way ANOVA. Source data are available online for this figure.

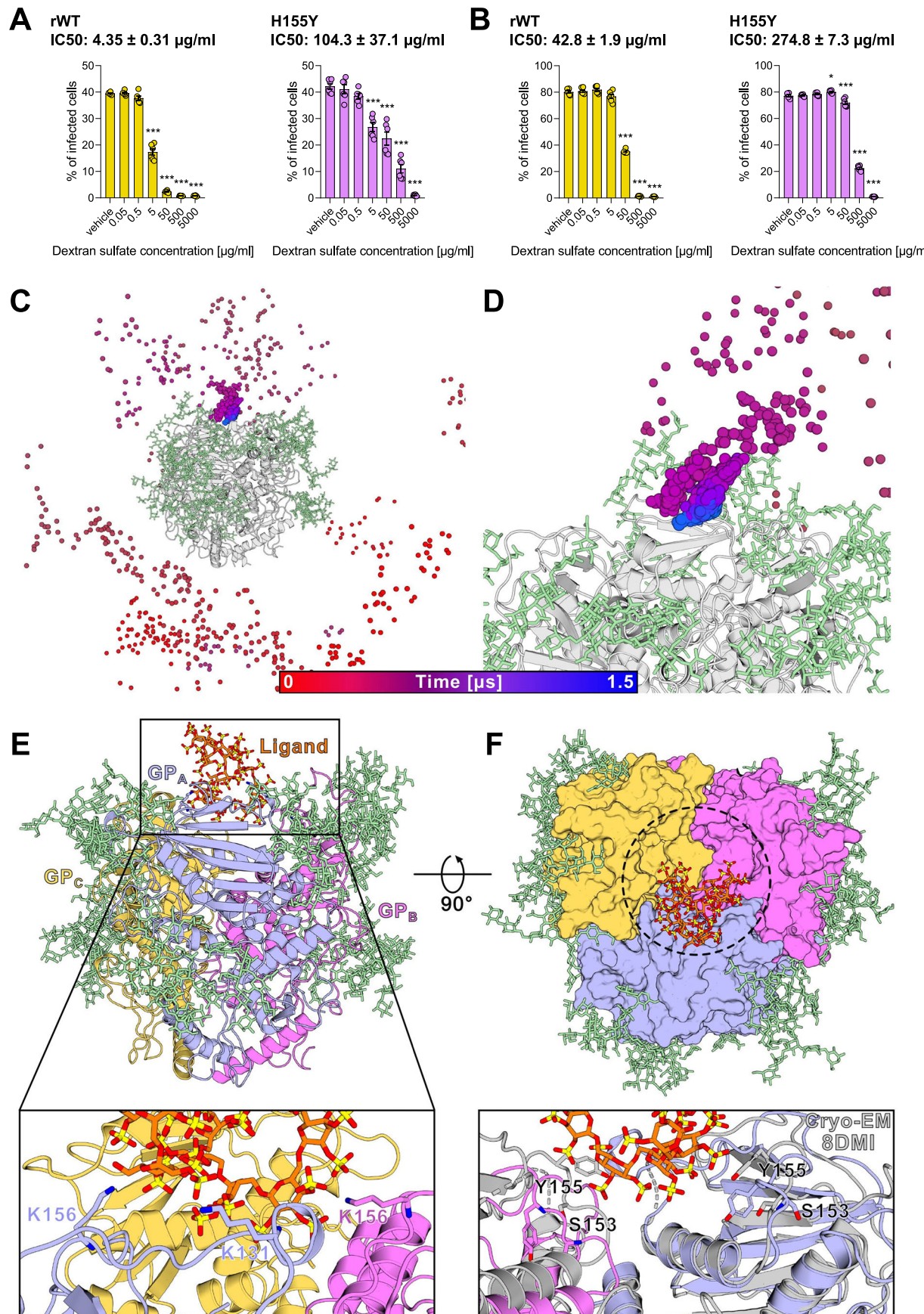

◄ **Figure 4. Sulfated dextran binds spontaneously to LCMV GP in unbiased molecular dynamics simulations.**

(A, B) MC57G (A) and BHK (B) cells were treated with dextran sulfate for 1 h. Next, cells were incubated with rWT (yellow) or H155Y (purple) viruses (MOI 0.5) for 1 h at 4 °C. To initiate infection, cells were moved to 37 °C for 24 h before adding monensin to block the Golgi apparatus and intracellular protein transport. Cells were then stained for viability and LCMV NP according to the standard protocol. Data presented as mean ± SEM, $n = 6$. ***$P < 0.001$ compared to vehicle control. Statistical significance was assessed by one-way ANOVA. (C) The path taken by the sulfated dextran molecule during simulation 6, visualized as spheres color-coded by the simulation time (gradient: red [0 μs] to blue [1.5 μs]). Each sphere represents the center of mass (COM) position of the ligand for every fifth trajectory frame, overlaid on the protein's average conformation (gray cartoon). (D) Close-up view of the predicted binding region from (C) (see (C) regarding the color scheme). (E) Representative structure of the sulfated dextran molecule (stick representation, orange) bound to the basic helix-loop face of monomer A (GP$_A$, blue) and B (GP$_B$, pink). The protein is shown with a cartoon representation, while glycans (green) are shown with stick representation. Close-up view of the simulated sulfated dextran binding region with interacting key residues (Lys131, Lys156 of GP$_A$ and Lys156 of GP$_B$) depicted as sticks. (F) Surface representation of the trimeric LCMV GP complex (top view). The surface-exposed cleft, which is partially occupied by the ligand, is indicated with a dotted circle. Close-up view of the simulated binding pose superimposed on the LCMV GP cryo-EM structure (gray cartoon, PDB ID: 8DMI), highlighting the conformational changes of Ser153 and Tyr155. Data were taken from simulation 6, which is also shown in Appendix Fig. S5B. Source data are available online for this figure.

bound to the GP via unspecific carbohydrate-mediated interactions with the surface glycans (Appendix Fig. S5A). Similar binding events were also observed in the simulations with the sulfated analog; however, in 2 of the 10 simulations, the sulfated dextran molecule bound to the helix-loop face of the N-terminal receptor binding domain, GP1 (Appendix Fig. S5B). This site has a largely basic surface with positively charged residues (Hastie et al, 2016) that engage with the sulfated dextran. In one of these two simulations, the sulfated dextran entered the center of the trimeric interface (Fig. 4C,D), where the major α-DG recognition determinant residues (Ser153, Tyr155, Arg190, and Leu260) are located (Hastie et al, 2016; Sevilla et al, 2000; Smelt et al, 2001). In this binding mode, the negatively charged sulfate groups of the ligand interacted with Lys131 and Lys156 of monomer A, as well as Lys156 of monomer B (Fig. 4E). In addition, upon binding of the sulfated dextran, loop 1 of monomers A and B underwent conformational changes, in which Ser153 and Tyr155 were facing downward the central cleft, shielded by the ligand (Fig. 4F). These simulations strengthen our previous conclusions about inhibition of viral entry mediated by binding of dextran sulfate to the virus, particularly to viral GP. Further, molecular modeling pointed to interactions of dextran sulfate with α-DG-specific amino acid moieties, suggesting direct inhibition of the virus binding to its cell surface receptor. Our in vitro data also support this hypothesis, since LCMV variants with high affinity to α-DG were less potently inhibited by dextran sulfate.

## Administration of dextran sulfate during the early stage of LCMV infection leads to impaired T-cell responses, virus persistence, and pathology

Next, we wondered whether in vivo application of dextran sulfate can modulate Arenavirus infection and the consequent immune activation. First, LCMV WE particles were pretreated with dextran sulfate, and subsequently injected i.v. into C57BL/6J mice. Mice infected with the non-pretreated virus served as a control group. At 6 h p.i., the first group of mice was injected again with the same dose of dextran sulfate, while control mice were injected with the adequate volume of PBS (Fig. 5A). As expected, LCMV titers in spleen and liver tissue from animals that received dextran sulfate treatment were reduced when compared to organs from control animals (Fig. 5B). Analysis of snap-frozen spleen sections for virus-infected cells by coimmunostaining for LCMV NP (clone VL4), CD169, and F4/80 (Fig. 5C), revealed an increased frequency of

infected cells in control animals (Fig. 5D), particularly CD169-positive macrophages (Fig. 5E), without any changes in the abundance of CD169$^+$ cells (Fig. 5F). Similar tendency was observed while analyzing snap-frozen liver sections with decreased expression of LCMV NP in dextran sulfate-treated group (Appendix Fig. S6A,B). It is worth noting that at this stage, dextran sulfate did not cause liver damage, as indicated by the activity of aspartate transaminase (AST) and alanine transaminase (ALT), well-known markers of liver damage, in serum samples (Appendix Fig. S6C). Viral replication in splenic CD169$^+$ cells can promote innate and adaptive immune activation, including production of type I interferons (Honke et al, 2012; Xu et al, 2015; Shaabani et al, 2016; Shinde et al, 2018; Casella et al, 2023). Accordingly, on day 1 p.i., decreased IFN-α levels were detected in the serum of animals treated with dextran sulfate when compared to controls (Fig. 5G). Consistently, reduced expression of interferon-stimulated genes (ISGs) and viral transcripts after dextran sulfate treatment was also determined in spleen and liver samples (Fig. 5H).

Next, we wondered whether this inhibition of infection may influence adaptive immune activation. We observed inhibition of rWT and H155Y virus entry following treatment of bone marrow-derived dendritic cells (BMDCs) with dextran sulfate (Fig. 6A). Accordingly, the expression of surface molecules involved in antigen presentation and co-stimulation was reduced on BMDCs treated with dextran sulfate prior to infection (Fig. 6B). This in turn led to limited expression of markers associated with proliferation (Figs. 6C and EV2A,C) and differentiation (Fig. EV2B), as well as IFN-γ production (Fig. 6D) by LCMV-specific CD8$^+$ P14$^+$ T cells co-cultured with dextran sulfate-treated and LCMV-infected BMDCs. Notably, when we treated CD8$^+$ T cells with physiologically-relevant concentrations of dextran sulfate in vitro, we measured a slight but significant increase in T-cell proliferation and surface marker expression associated with effector differentiation, but did not observe impaired T-cell proliferation in this setting (Fig. EV2D,E). Next, we investigated whether limited replication and immune activation during early infection affected T-cell immunity and the outcome after infection in vivo (Fig. 6E). For this purpose, we selected two time points, 8 and 12 days p.i. with LCMV WE. In this experimental setting, the peak phase of active infection is observed at day 8 p.i., while by day 12 p.i., the virus is usually cleared in spleen and liver tissue. We observed a decreased number of antiviral CD8$^+$ T cells specific for the immunodominant epitope gp33 and np396 in animals treated with dextran sulfate in the blood on day 8 p.i. (Fig. EV3A), and in blood,

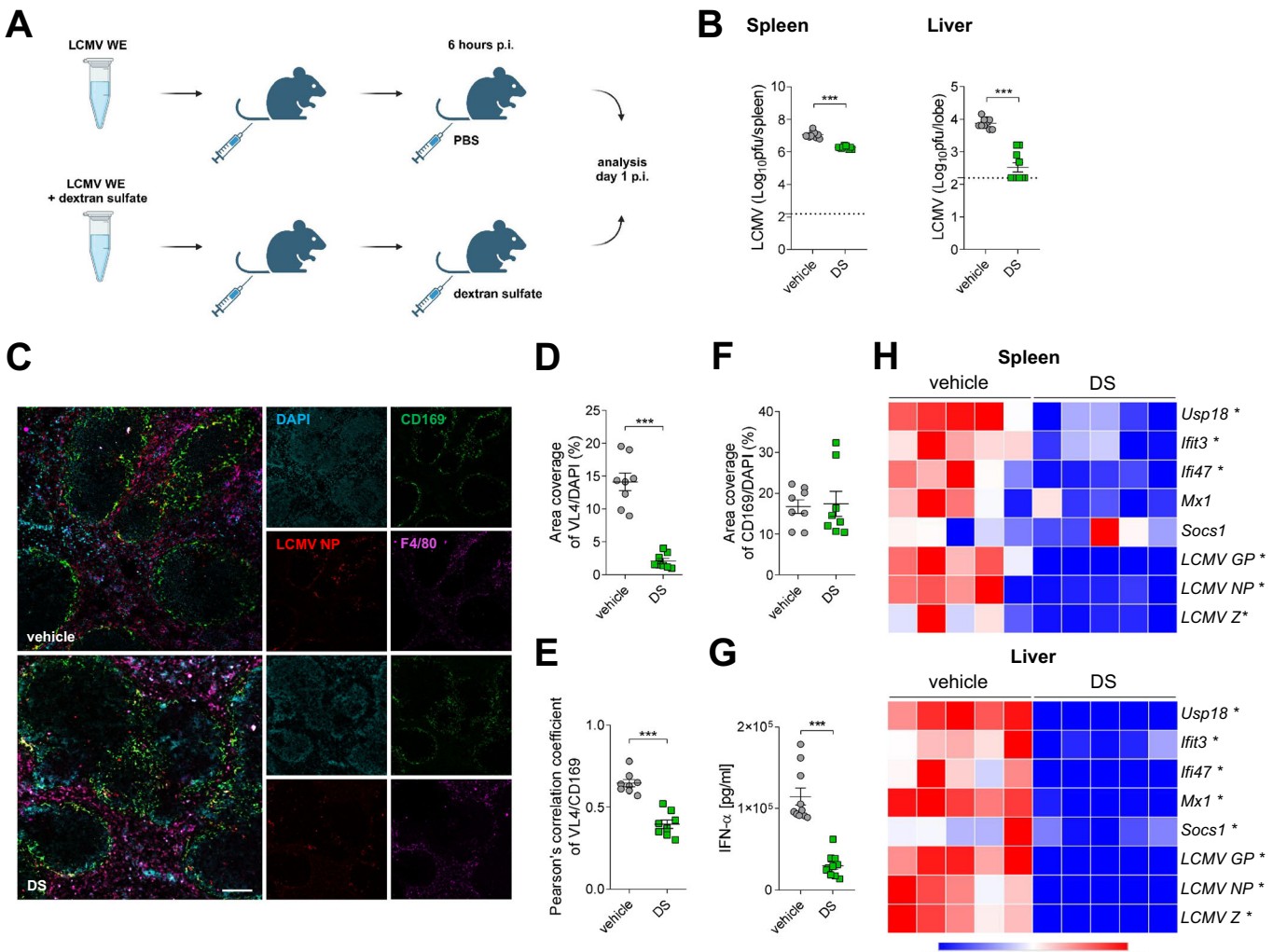

**Figure 5. Dextran sulfate decreases IFN production and viral titers during early stages of infection.**

LCMV WE ($5 \times 10^6$ pfu) was pretreated with dextran sulfate (DS, 500 µg/ml) for 30 min, and then injected i.v. into C57BL/6J mice ($10^6$ pfu per mouse), while control mice were infected with LCMV WE. Mice were re-injected i.v. with dextran sulfate (100 µg per mouse) 6 h p.i., and control mice were injected with PBS. Analyses were carried out on day 1 p.i. (A) Scheme of experimental setup. (B) LCMV titers were determined in spleen and liver tissue collected on day 1 p.i. Data presented as mean ± SEM, $n = 10$ mice per condition, ***$P < 0.001$. Statistical significance was assessed by Student's $t$ test. (C–F) Spleen tissue samples were stained for LCMV NP (clone VL4), CD169, and F4/80 (C); representative pictures of $n = 4$ mice per condition are shown (ROI = 2), scale bar = 100 µm. LCMV NP (D) and CD169 (F) staining normalized to DAPI, and colocalization of LCMV NP and CD169 in spleen tissue (E) was quantified. Data presented as mean ± SEM, ***$P < 0.001$. Statistical significance was assessed by Student's $t$ test. Analysis performed in ImageJ, with additional use of JACoP co-localization plugin. (G) IFN-α levels were determined in serum samples. Data presented as mean ± SEM, $n = 10$ mice per condition, ***$P < 0.001$. Statistical significance was assessed by Student's $t$ test. (H) mRNA expression of ISGs and LCMV genes was determined in spleen and liver tissue as indicated. Data presented as heatmaps representing the number of cognate mRNA copies per copy of mRNA for the reference housekeeping gene, $n = 5$ mice per condition. *$P < 0.05$ between control and DS-treated groups. Statistical significance was assessed by Student's $t$ test. Source data are available online for this figure.

spleen and liver on day 12 p.i. when compared to controls (Fig. 6F). Within these populations, the effector T-cell subsets (memory precursor and short-lived effector cells (MPECs and SLECs), defined as KLRG1⁻ IL7R⁺ and KLRG1⁺ IL7R⁻, respectively) were also significantly reduced after dextran sulfate treatment (Figs. 6G and EV3B). Additional staining for T-cell exhaustion makers revealed that LCMV-specific T cells expressed increased levels of surface PD-1, TIM-3, 2B4, and LAG3 molecules in samples collected from animals treated with dextran sulfate (Fig. EV3C–F). To test the functionality of the CD8⁺ T cells, blood samples and the single-cell suspensions from spleen and liver were restimulated

ex vivo with LCMV gp33-41 and np396-404 peptides. Dextran sulfate treatment decreased the number of IFN-γ- and TNF-α-producing CD8⁺ T cells in blood on day 8 p.i. (Fig. EV3G) and IFN-γ-producing CD8⁺ T cells in blood, spleen, and liver on day 12 p.i. (Fig. 6H). The observed impaired T cell functions in samples from mice exposed to dextran sulfate were accompanied by increased liver damage, determined by analysis of ALT/AST activity in serum on day 8 (Fig. EV3H) and day 12 p.i. (Fig. 6I). Furthermore, we observed higher viral titers in blood and organs in comparison to control animals (Figs. 6J and EV3I). Viral load in organs was additionally confirmed by analysis of viral transcripts

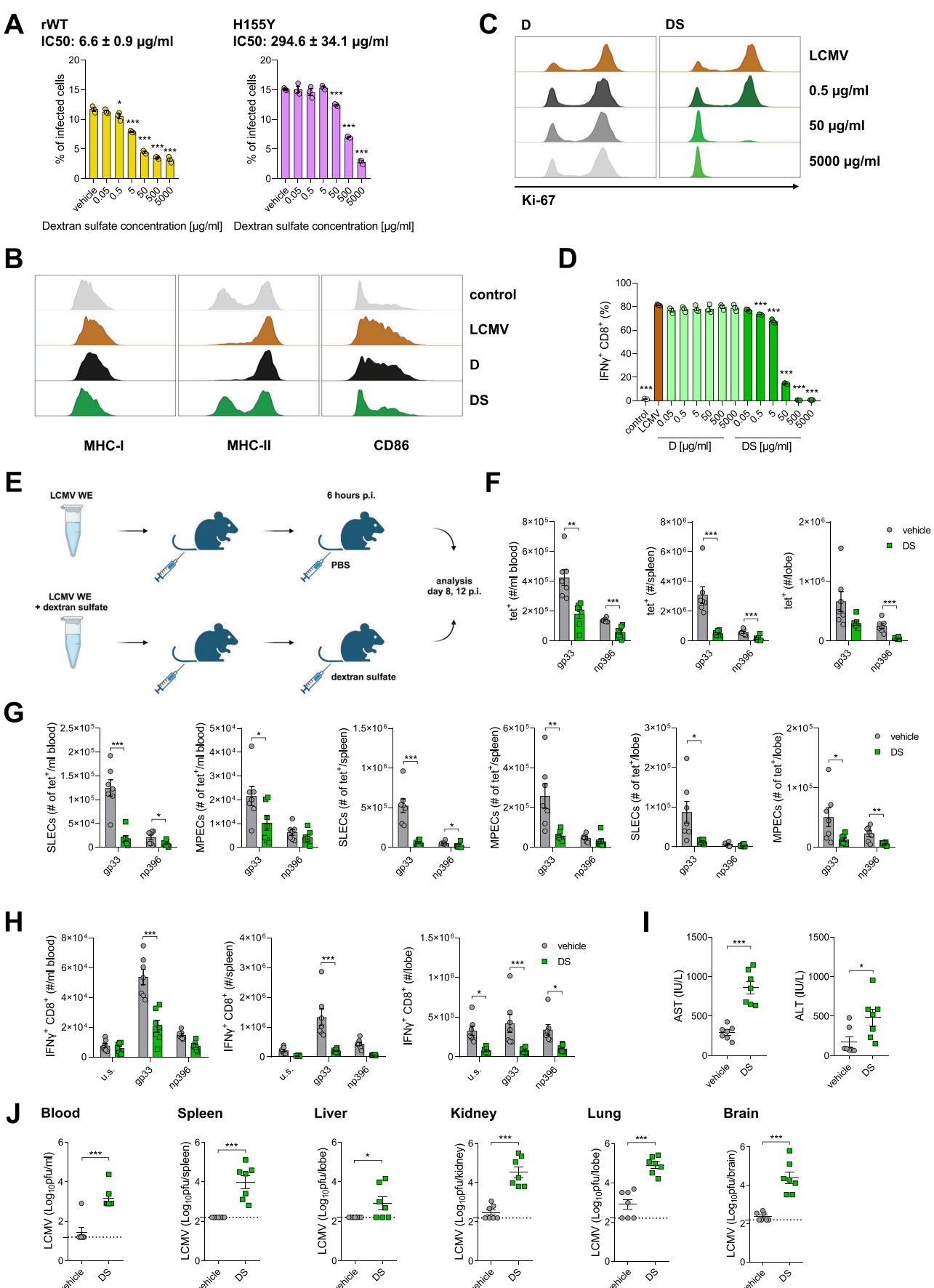

**Figure 6.  Treatment with dextran sulfate at the beginning of infection leads to virus persistence and pathology.**

(A) BMDCs were treated with dextran sulfate for 1 h at 4 °C. Next, cells were incubated with rWT and H155Y viruses (MOI 0.5) for 1 h at 4 °C. To initiate infection, cells were moved to 37 °C for 24 h before adding monensin to block the Golgi apparatus and intracellular protein transport. Cells were then stained for viability and LCMV NP according to the standard protocol. Data presented as mean ± SEM, $n = 3$. *$P < 0.05$, ***$P < 0.001$ compared to vehicle control. Statistical significance was assessed by one-way ANOVA. (B) BMDCs were treated with dextran/dextran sulfate for 1 h. Cells were subsequently infected with LCMV WE (MOI 0.5) for 1 h. Then the dextrans and virus were removed by washing. Expression of surface molecules was analyzed after 24 h on CD11c$^+$ cells as indicated. Representative histograms (for dextran (D) and dextran sulfate (DS) at 500 μg/ml) showing surface molecule expression on BMDCs are shown ($n = 3$). (C, D) BMDCs were treated with dextran/dextran sulfate for 1 h. Cells were subsequently infected with LCMV WE (MOI 0.5) for 1 h. Then the dextrans and virus were removed by washing. CD8$^+$ P14$^+$ T cells were isolated and mixed with BMDCs at a ratio of 10:1. After 72 h of incubation, T cells were analyzed for Ki-67 expression (C) and IFN-γ production (D). Representative histograms showing Ki-67 expression in BMDCs are shown ($n = 3$). For IFN-γ production, data presented as mean ± SEM, $n = 3$. ***$P < 0.001$ compared to LCMV-infected control. Statistical significance was assessed by one-way ANOVA. (E–J) LCMV WE ($5 \times 10^6$ pfu) was pretreated with dextran sulfate (DS, 500 μg/ml) for 30 min, and then injected i.v. into C57BL/6J mice ($10^6$ pfu per mouse), while control mice were infected with LCMV WE. Mice were re-injected i.v. with dextran sulfate (100 μg per mouse) 6 h p.i., and control mice were injected with PBS. Analyses were carried out on days 8 and 12 p.i. (E) Scheme of experimental setup. (F) Tet-gp33$^+$ and tet-np396$^+$ T cells were determined in the blood, spleen, and liver. (G) SLECs (IL7R$^-$, KLRG1$^+$) and MPECs (IL7R$^+$, KLRG1$^-$) subsets within tet$^+$ T cells were determined in blood, spleen, and liver. (H) IFN-γ production by CD8$^+$ T cells in blood, spleen, and liver after re-stimulation with LCMV-specific peptides was determined. (I) ALT and AST activity in the serum of control and DS-treated mice was evaluated. (J) LCMV titers were determined in blood, spleen, liver, kidney, lung, and brain. Tet$^+$ CD8$^+$ T cells, SLECs, MPECs (both populations as subsets of tet$^+$ cells), and cytokine-producing CD8$^+$ T cells are presented as absolute counts: number of cells per ml of blood, per spleen, or per liver lobe, as indicated. Data presented as mean ± SEM, $n = 7$ mice per condition, *$P < 0.05$, **$P < 0.01$, ***$P < 0.001$. Statistical significance was assessed by Student's *t* test, or two-way ANOVA in the case of IFN-γ production. Source data are available online for this figure.

via RT-PCR (Fig. EV3J). Taken together, these data show that the treatment with dextran sulfate at the beginning of infection causes reduced viral replication, resulting in impaired immune activation, including reduced T-cell function, leading to prolonged viral infection and pathology.

## Dextran sulfate treatment in the course of active LCMV infection enhances antiviral immune responses and viral clearance

Considering the possibility to inhibit LCMV replication, we hypothesized that administration of dextran sulfate during the active phase of infection may limit viral spread without affecting antigen presentation and T-cell activity during earlier stages. Thus, we infected C57BL/6J mice with LCMV WE i.v. and treated one group with dextran sulfate on days 6 and 7 p.i. At the same time, control animals were injected with PBS. The T-cell function and viral persistence were evaluated on day 8 p.i. (Fig. 7A), in order to investigate the direct and immediate effects of dextran sulfate administration. In this experimental setup, an increased number of gp33- and np396-specific CD8$^+$ T cells was observed in the blood of animals treated with dextran sulfate (Fig. 7B), with slightly increased numbers of MPECs and SLECs (Fig. 7C). The same parameters in spleen and liver remained unchanged (Fig. EV4A,B). Dextran sulfate treatment significantly increased the number of IFN-γ- and TNF-α-producing CD8$^+$ T cells in blood (Fig. 7D), and only slightly in liver, where higher numbers of IFN-γ-producing CD8$^+$ T were observed in dextran sulfate-treated animals (Fig. EV4C). LCMV-specific T cells expressed decreased levels of exhaustion markers in blood and liver samples collected from animals treated with dextran sulfate, but no differences were observed in spleen samples (Fig. EV4D–F). Administration of dextran sulfate in this experimental setup led to decreased viral loads in spleen and liver, as determined by the level of viral titer and LCMV RNA (Fig. 7E,F). At the same time, viral titers in blood, kidney, lung, and brain tissue remained unchanged (Fig. EV4G). Coimmunostaining of snap-frozen liver sections envisaged a significant decrease in expression of LCMV NP in dextran sulfate-treated animals (Fig. 7G,H). Reduced levels of LCMV NP

were also detected in snap-frozen spleen sections (Fig. 7I,J). Moreover, we observed reduced expression of collagen in snap-frozen liver tissue of dextran sulfate-treated animals when compared to control mice (Fig. 7K). Consistently, the activities of ALT and AST were reduced in mice receiving dextran sulfate when compared to control mice (Fig. 7L). Of note, when we performed a similar in vivo experiment using LCMV clone 13, we observed increased cytokine production and lowered viral titers in the blood of mice treated with dextran sulfate on day 8 p.i. This effect however did not persist in time, and on day 20 p.i. we did not observe any differences in antiviral T-cell immunity or organ titers (Fig. EV5). Since LCMV clone 13 is a high-affinity virus, we speculate that repeated administration of dextran sulfate over a longer period of time could potentially contribute to viral clearance.

## Discussion

In this study, we identify sulfated glycosaminoglycans (GAGs) as potent inhibitors of Arenavirus cell entry, directly blocking viral attachment to host cells. Extending previous reports on this topic, we demonstrate in vivo antiviral activity of dextran sulfate and show that treatment timing critically determines the outcome of LCMV infection: early administration reduced infection of immune cells and decreased innate activation, leading to impaired T-cell priming and prolonged viral persistence. By contrast, treatment during the active phase of infection decreased viral titers and enhanced antiviral immunity. These findings uncover a previously unrecognized dual role of GAGs in modulating Arenavirus infection and host immunity, and establish sulfated polysaccharides as promising candidates for targeting virus-host attachment interfaces.

Arenaviruses enter the host cells primarily through receptor-mediated endocytosis. α-dystroglycan (α-DG) is the main cellular receptor used for infection by Old World Arenaviruses (including LASV and LCMV), although in the absence of proper α-DG glycosylation, they may also use members of the Tyro3/Axl/Mer (TAM) protein family, DC-SIGN, LSECtin (Pasqual et al, 2011; Shimojima et al, 2012; Shimojima and Kawaoka, 2012; Fedeli et al,

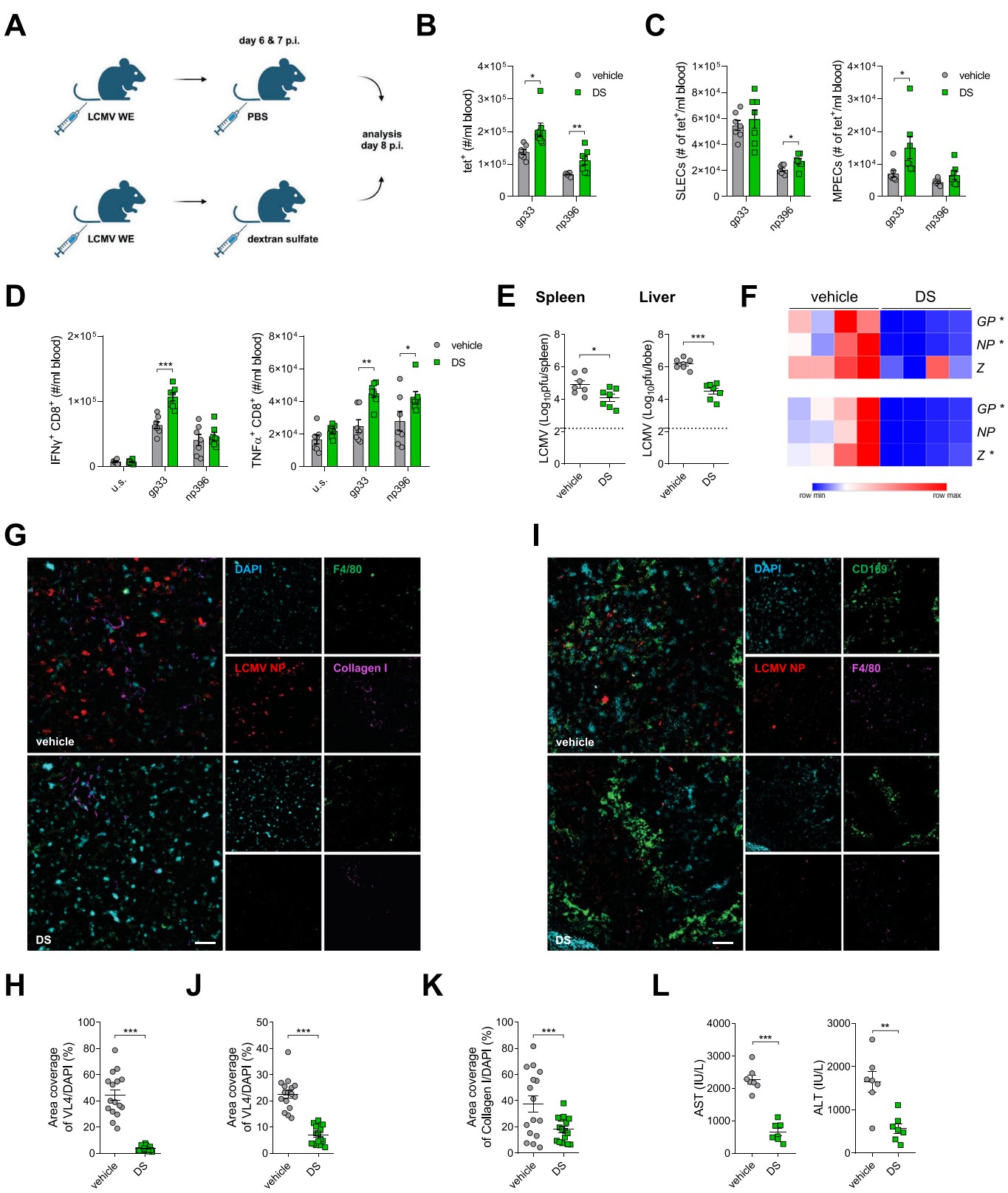

**Figure 7.  Treatment with dextran sulfate in the course of acute infection leads to increased antiviral immunity.**

C57BL/6J mice were infected i.v. with LCMV WE ($10^6$ pfu per mouse). On days 6 and 7 p.i., mice were injected i.v. with dextran sulfate (DS, 100 μg per mouse), while control mice were injected with PBS. Analyses were carried out on day 8 p.i. (A) Scheme of experimental setup. (B) Tet-gp33[+] and tet-np396[+] T cells were determined in blood. (C) SLECs (IL7R[-], KLRG1[+]) and MPECs (IL7R[+], KLRG1[-]) subsets within tet[+] T cells were determined in blood. (D) IFN-γ and TNF-α production by CD8[+] T cells in blood after re-stimulation with LCMV-specific peptides was determined. (E) LCMV titers were determined in spleen and liver tissue. Tet[+] CD8[+] T cells, SLECs, MPECs (both populations as subsets of tet[+] cells), and cytokine-producing CD8[+] T cells are presented as absolute counts: number of cells per ml of blood, as indicated. Data presented as mean ± SEM, $n = 7$ mice per condition, *$P < 0.05$, **$P < 0.01$, ***$P < 0.001$. Statistical significance was assessed by Student's *t* test, or two-way ANOVA in the case of IFN-γ and TNF-α production. (F) mRNA expression of LCMV genes was determined in the spleen (upper panel) and liver (lower panel). Data presented as heatmaps representing the number of cognate mRNA copies per copy of mRNA for the reference housekeeping gene, $n = 4$ mice per condition. *$P < 0.05$ between control and DS-treated groups. Statistical significance was assessed by Student's *t* test. (G, H) Liver tissue samples were stained for LCMV NP (clone VL4), F4/80, and collagen I (G); representative pictures of $n = 4$ mice per condition are shown (ROI = 4), scale bar = 50 μm. LCMV NP (H) staining normalized to DAPI was quantified. Data presented as mean ± SEM, ***$P < 0.001$. Statistical significance was assessed by Student's *t* test. Analysis performed in ImageJ. (I, J) Spleen tissue samples were stained for LCMV NP (clone VL4), CD169, and F4/80 (I); representative pictures are shown, $n = 4$ mice per condition (ROI = 4), scale bar = 50 μm. LCMV NP staining in spleen normalized to DAPI (J) was quantified. Data presented as mean ± SEM, ***$P < 0.001$. Statistical significance was assessed by Student's *t* test. Analysis performed in ImageJ. (K) Collagen I staining normalized to DAPI was quantified in liver tissue samples. Data presented as mean ± SEM, ***$P < 0.001$. Statistical significance was assessed by Student's *t* test. Analysis performed in ImageJ. (L) ALT and AST activity in the serum of control and DS-treated mice was evaluated. Data presented as mean ± SEM, $n = 7$ mice per condition, **$P < 0.01$, ***$P < 0.001$. Statistical significance was assessed by Student's *t* test. Source data are available online for this figure.

2018), and in specific cases, heparan sulfate proteoglycans (HSPGs) (Jae et al, 2014; Fedeli et al, 2018; Volland et al, 2021). On the other hand, New World Arenaviruses (such as Junin, Machupo, or Guanarito) exploit transferrin receptor 1 (TfR1) for cell entry (Radoshitzky et al, 2008; Zong et al, 2014).

Here, we speculate that glycosaminoglycan-mediated blocking of cellular binding and entry could be a potential strategy for the treatment of arenaviral infections. Soluble heparin, chondroitin sulfate, and protamine sulfate have been shown to inhibit infection of vesicular stomatitis virus (VSV) variants carrying LCMV GP with low affinity to α-DG (Volland et al, 2021). On the other hand, a high-affinity LCMV GP variant was found not to interact with heparin, consistent with previous reports (Kunz et al, 2001; Lee et al, 2008). Further analyses showed that posttranslational modification of surface proteoglycans in the Golgi compartment plays a major role in efficient infection of all tested VSV GP constructs, especially for low-affinity variants and in the absence of functional α-DG (Volland et al, 2021). However, the mechanistic and structural basis, as well as the in vivo relevance of these observations, remained unclear.

Our findings extend this body of knowledge by demonstrating that highly sulfated GAGs, particularly dextran sulfate, are capable of blocking Arenavirus-cell binding and entry, and importantly, that this activity may contribute to reduced viral burden and pathology in vivo. Analyzing the inhibitory activity of GAGs towards both pathogenic and non-pathogenic Arenaviruses, we observed dextran sulfate to be more potent than heparin, which might be based on the higher content of negatively charged sulfate groups (Kjellén and Lindahl, 2018; Hao et al, 2021; Andreu et al, 2023). It is worth noting here that, depending on the species of the virus, a shift in the range of effective concentrations was observed. We speculate that these changes may be attributed to the differences in amino acid sequence of the GPs across Arenavirus species (Abraham et al, 2010; Martin et al, 2010; Moon-Walker et al, 2023) that can cause variation in GAG binding strength, and also different affinity to entry receptors on host cells. Consistently, we found that inhibition of viral entry of high-affinity GP-expressing LCMV required higher concentrations of dextran sulfate. Further, it has been shown that even a single amino acid mutation in viral protein may significantly influence the binding of

GAGs (Silva et al, 2014), which is also true in the case of LCMV strains with different affinities to α-DG due to the point mutations in LCMV GP (Kunz et al, 2001; Lee et al, 2008; Volland et al, 2021). Thus, we speculate that dextran sulfate has potent inhibitory activity, potentially via binding specific domains on the LCMV GP. Indeed, our molecular dynamics simulations revealed that dextran sulfate binds LCMV GP predominantly due to the interactions of negatively charged sulfate moieties with positively charged lysine residues located in the receptor-binding cleft. These observations suggest that sulfated GAGs inhibit viral receptor binding and subsequent cell entry by direct interaction with viral glycoproteins. Further, these results provide mechanistic insight into how sulfated glycans modulate viral tropism and suggest that charge-based interference with viral receptor engagement could form the basis for broad-spectrum antiviral design. Future studies should focus on developing synthetic sulfated glycan analogs with optimized charge distribution and safety profiles, and on further structural studies to determine the specific binding determinants within Arenavirus GP.

An equally important issue concerns the strategy of administering sulfated GAGs during infection. Our research indicates that in contrast to the treatment during the active phase of infection, early administration may lead to insufficient infection of antigen-presenting cells, resulting in limited T-cell activation and prolonged viral persistence, pointing to the critical role of treatment timing in achieving therapeutic efficacy.

In conclusion, our study demonstrates that the highly sulfated nature of GAGs allows them to limit arenaviral infections. This antiviral effect is likely mediated by electrostatic interactions between the negatively charged sulfated GAGs and positively charged regions on the viral envelope proteins, preventing their attachment to cell surface receptors. Among the tested compounds, dextran sulfate emerged as the most potent antiviral GAG selected during in vitro screening, effectively inhibiting LCMV cell entry during early stages of infection. Additionally, it exhibited antiviral activity in vivo when administered during the active phase of infection. These findings highlight the therapeutic potential of sulfated GAGs as novel agents for treating Arenavirus infections, and provide valuable insights into virus-host interactions and the development of carbohydrate-based antiviral therapies.

# Methods

### Reagents and tools table

| Reagent/resource | Reference or source | Identifier or catalog number |
| --- | --- | --- |
| **Experimental models** | | |
| C57BL/6J (M. musculus) | Jacksons Laboratory | N/A |
| CD45.1 + P14+ (M. musculus) | Prof. Pamela Ohashi, University of Toronto, Toronto, Canada | N/A |
| MC57G | ATCC | CRL-2295 |
| BHK | ATCC | CCL-10 |
| Vero | ATCC | CCL-81 |
| HEK293T | ATCC | CRL-3216 |
| HEK293T α-DG ko | https://doi.org/10.1128/JVI.00093-18 | N/A |
| **Antibodies** | | |
| Anti-CD169 | Bio-Rad | MCA884F |
| Anti-F4/80 | Invitrogen | 45-4801-82 |
| Anti-CD31 | Invitrogen | 11-0311-82 |
| Anti-Collagen I | Abcam | AB309367 |
| Anti-CD8 | Invitrogen | 46-0081-82 |
| Anti-CD8 | Invitrogen | 11-0081-85 |
| Anti-CD19 | Invitrogen | 47-0193-82 |
| Anti-2B4 | Invitrogen | 25-2441-82 |
| Anti-IL7R | Invitrogen | 11-1271-85 |
| Anti-KLRG1 | Invitrogen | 48-5893-82 |
| Anti-TIM-3 | Invitrogen | 12-5870-82 |
| Anti-CD44 | BD Biosciences | 563736 |
| Anti-CD62L | BD Biosciences | 563117 |
| Anti-PD-1 | BD Biosciences | 562523 |
| Anti-LAG3 | BD Biosciences | 563179 |
| Anti-MHC-I | Invitrogen | 1-5999-85 |
| Anti-MHC-II | Invitrogen | 56-5321-82 |
| Anti-CD86 | BD Biosciences | 563077 |
| Anti-IFNg | Invitrogen | 17-7311-82 |
| Anti-TNFa | Invitrogen | 12-7321-81 |
| Anti-Ki67 | Invitrogen | 56-5698-82 |
| Anti-CD3 | Invitrogen | 16-0031-86 |
| Anti-CD28 | BD Biosciences | 553294 |
| Anti-LCMV NP | https://doi.org/10.1038/ncomms14447 | clone: VL4 |
| Anti-LCMV GP | https://doi.org/10.1038/ncomms14447 | clone: KL25 |
| Peroxidase AffiniPure® Goat Anti-Rat IgG | Jackson Immunoresearch | 112-035-003 |
| Goat anti-Human IgG Fc Secondary Antibody | Invitrogen | 12-4998-82 |
| Goat Anti-Mouse IgG H&L (HRP) | Abcam | ab6789 |

| Reagent/resource | Reference or source | Identifier or catalog number |
| --- | --- | --- |
| **Oligonucleotides and other sequence-based reagents** | | |
| PCR primers | This study | Table 1 |
| **Chemicals, enzymes, and other reagents** | | |
| DMEM | PAN BIOTECH | P04-03600 |
| αMEM | PAN BIOTECH | P04-21150 |
| RPMI 1640 | PAN BIOTECH | P04-17525 |
| HBSS | Sigma-Aldrich | H9269 |
| FBS | Gibco | A5256701 |
| PSG | Gibco | 10378-016 |
| β-mercaptoethanol | Sigma-Aldrich | M6250 |
| Monensin | Invitrogen | 00-4505-51 |
| Dapi | Invitrogen | Cat#D1306 |
| Resazurin | Sigma-Aldrich | R7017 |
| Brefeldin A | Invitrogen | 00-4506-51 |
| Saponin | Sigma-Aldrich | S4521 |
| GM-CSF | PAN BIOTECH | Cat#CB-2210002 |
| Cell proliferation dye eFluor 670 | Invitrogen | 65-0840-85 |
| Fixable viability dye eFluor 450 | Invitrogen | 65-0863-14 |
| PE-secondary goat anti-rat IgG (entry assay) | Jackson Immunoresearch | 112-116-072 |
| Annexin V | ImmunoTools | Cat#31490016 |
| 7AAD | Invitrogen | Cat#00-6993-50 |
| Annexin Binding Buffer | BD Biosciences | 51-66121E |
| Vectashield Vibrance Antifade mounting medium | Vector Labs | H-1700 |
| Mouse CD8 purification kit | Miltenyi Biotec | 130-104-075 |
| RNeasy Kit for RNA isolation | QIAGEN | Cat#74106 |
| iTaq Universal SYBR Green 1-step kit | Bio-Rad | Cat#1725151 |
| Mouse IFN-α ELISA kit | Invitrogen | Cat#BMS6027TEN |
| Cu Grids 200 M Pioloform | Plano | SP162 |
| Diatome Diamond Knife, ultra 45° | Diatome | N/A |
| Amicon Ultra centrifugal filter device | Merck Millipore | UFC510096 |
| Spotchem Liver-1 | Arkray | 77182 |
| Hydrochloric acid 1 M | Sigma-Aldrich | 1101652500 |
| Diethyl ether (p.a.) | Sigma-Aldrich | 32203-1L-M |
| Dichloromethane (99.9%, puriss., p.a.) | Sigma-Aldrich | 32222-2.5L-M |
| D-(+)-mannose (99%) | Sigma-Aldrich | M8574-100G |
| Boron trifluoride diethyl etherate (for synthesis) | Sigma-Aldrich | 8016470250 |
| Sodium acetate | Sigma-Aldrich | 241245-500 G |
| Dichloromethane (p.a.) | ACROS Organics | 390700025 |

| Reagent/resource | Reference or source | Identifier or catalog number |
|---|---|---|
| Dimethylformamide (98%, for peptide synthesis) | ACROS Organics | 354830025 |
| Sulfur trioxide trimethylamine | ACROS Organics | 159351000 |
| Ethyl acetate (analytical reagent grade) | ACROS Organics | 232110025 |
| Methanol (p.a.) | VWR Chemicals | 8.22283.2500 |
| Acetic anhydride (99.7%) | VWR Chemicals | 222130010 |
| Pyridine | VWR Chemicals | 27194.295 |
| Diphenyl-(2,4,6-trimethylbenzoyl)-phosphine oxide ( > 98%) | TCI chemicals | D3358 |
| N-hydroxyethylacrylamide ( > 98%) | TCI chemicals | H1262 |
| Magnesium sulfate (dried, laboratory reagent grade) | Fisher Scientific | 10776101 |
| Sodium methoxide (98%) | Fisher Scientific | 11377535 |
| Formaldehyde solution | Sigma-Aldrich | 252549 |
| TritonX-100 | Sigma-Aldrich | T9284 |
| Carboxymethylcellulose | Calbiochem | 217274 |
| Tetramethylbenzidine | Sigma-Aldrich | 860336 |
| Acetone | Sigma-Aldrich | 320110 |
| Cacodylic acid | Serva | 15540 |
| Chloroform | Merck | 1.02445 |
| DER | Serva | 18247.01 |
| ERL | Serva | 21041.02 |
| Ethanol | Merck | 1.00983 |
| Glutaraldehyde 25% high purity | Serva | 23114 |
| Lead(II) nitrate | Merck | 1.07398 |
| NSA | Serva | 30812.01 |
| Osmium(VIII)oxide | EMS | 19110 |
| Paraformaldehyde | Merck | 1.04005 |
| Phosphotungstic acid | Merck | 1.00583 |
| DMAE (S1) | Serva | 20130 |
| Sodium hydroxide | Merck | 1.06498 |
| tri-Sodium citrate dihydrate | Merck | 1.06448 |
| Uranyl acetate | Merck | 8476 |
| **Software** | | |
| ImageJ | National Institutes of Health | https://imagej.net/ij |
| GraphPad Prism (v8.0.1) | GraphPad Software | graphpad.com |
| FlowJo (v10) | BD | flowjo.com |
| QuantiStudio Design & Analysis software (v2.7.0) | Thermo Fisher Scientific | thermofisher.com |
| **Other** | | |
| Monomer for tetramer-gp33 | NIH Tetramer Core Facility | N/A |

| Reagent/resource | Reference or source | Identifier or catalog number |
|---|---|---|
| Monomer for tetramer-np396 | NIH Tetramer Core Facility | N/A |
| GP33–41 KAVYNFATM peptide | Thermo Scientific | N/A |
| NP396–404 FQPQNGQFI peptide | JPT | N/A |

**Table 1. Primer sequences.**

| Gene | Forward and reverse sequences (5′–3′) |
|---|---|
| Gapdh | Fw: TGCACCACCAACTGC<br>Rv: GGATGCAGGGATGAT |
| Actb | Fw: GGCTGTATTCCCCTCCATCG<br>Rv: CCAGTTGGTAACAATGCCATGT |
| Usp18 | Fw: TGCCTCGGAGTGCAGAAGA<br>Rv: CGTGATCTGGTCCTTAGTCAGG |
| Ifit3 | Fw: CCTACATAAAGCACCTAGATGGC<br>Rv: ATGTGATAGTAGATCCAGGCGT |
| Ifi47 | Fw: CGAAGCAGATGAATCCGCTGA<br>Rv: TGCGTGGAAATTGGGTGTCC |
| Mx1 | Fw: GACCATAGGGGTCTTGACCAA<br>Rv: AGACTTGCTCTTTCTGAAAAGCC |
| Socs1 | Fw: CTGCGGCTTCTATTGGGGAC<br>Rv: AAAAGGCAGTCGAAGGTCTCG |
| LCMV NP | Fw: CCTAGATGAGTTGGCAACAA<br>Rv: GCTAAAGGATAAACACCCAGTTCTG |
| LCMV GP | Fw: ATCACGAGCATCAAAGCTGTGTA<br>Rv: TGAGAGTTGTTGGCTGAGCA |
| LCMV Z | Fw: TGACCGATCATCAGTCAT<br>Rv: TCTAACCAAGCTGTCAAATTTC |

## Cell culture

MC57G (mouse fibrosarcoma), Vero (African green monkey kidney fibroblasts), BHK (baby hamster kidney), and HEK293T (human embryonic kidney 293T) cell lines were purchased from American Type Culture Collection (ATCC). MC57G cells were cultured in Alpha MEM Eagle (PAN-Biotech) supplemented with 5% fetal bovine serum (Gibco) and 1% penicillin-streptomycin-glutamine (Thermo Fisher Scientific). HEK293T and BHK cells were cultured in DMEM (PAN-Biotech) supplemented with 10% fetal bovine serum (Gibco) and 1% penicillin-streptomycin-glutamine (ThermoFisher Scientific). Vero cells were grown in DMEM (PAN-Biotech) supplemented with 5% fetal bovine serum (Gibco), 1% penicillin-streptomycin-glutamine (PAN-Biotech), and 1% non-essential amino acids (PAN-Biotech). HEK293T α-DG ko cells were generated as previously described (Brouillette et al, 2018). Cells were maintained under standard conditions at 37 °C in an atmosphere of 5% $CO_2$. Cells were sub-cultured 2–3 times per week. Cells were not authenticated or tested for mycoplasma contamination during this study.

## Viruses

LCMV WE was originally obtained from F. Lehmann-Grube (Heinrich Pette Institute, Hamburg, Germany). LCMV clone 13 was obtained

from Max Löhning (German Rheumatology Research Center, Berlin, Germany). LASV strain Ba366 (Lecompte et al, 2006) was isolated in the Bernhard Nocht Institute for Tropical Medicine (BNITM). Lujo Virus (LUJV) strain SPU256/08 (Paweska et al, 2009) was obtained via the "EVA goes global" program (EVAg, H2020 grant no. 653316) from the National Institute for Communicable Diseases, A Division of the National Health Laboratory Service, Johannesburg, South Africa. Machupo Virus (MACV) strain Cavallo, and Junin virus (JUNV) strain XJ CL13 were obtained from Dr. Heinz Feldmann (Rocky Mountain Laboratories, Hamilton, USA). LASV, LUJV, MACV, and JUNV were handled in the biosafety level 4 laboratory at BNITM. The following viruses were handled in biosafety level 2 containment: Morogoro virus (MORV) strain 3017/2004 (Gunther et al, 2009) (isolated in BNITM), Tacaribe virus (TCRV) strain TRVL-11573 (obtained from Prof. Dr. Alison Groseth, Friedrich Löffler Institut, Riems, Germany), Parana virus (PARV) strain 12056 (obtained from Prof. Dr. Daniel Pinschewer, University of Basel, Basel, Switzerland).

LASV, LUJV, MACV, JUNV, MORV, TACV, and PARV were amplified in Vero cells. In short, cells were infected with an MOI of 0.01, and cell culture supernatant was harvested after 3 or 4 days. The supernatant was cleared of cell debris by centrifugation at 4000 × $g$ for 10 min, aliquoted, and stored at −80 °C. Virus titers were determined by immunofocus assay.

## Recombinant viruses

Recombinant WT (rWT) and H155Y viruses were created as described previously (Xu et al, 2021; Xu et al, 2024). Both viruses consist of the L segment from clone 13 strain and S segment from WE strain.

## Compounds

Heparin (catalog number: H4784), dextran (D9260), dextran sulfate 5 kDa (31404), dextran sulfate 9-20 kDa (D6924), chondroitin sulfate A (C9819), and chondroitin sulfate SC (C4384) were purchased from Sigma-Aldrich. Hyaluronic acid 15 kDa (Fh63426), hyaluronic acid 50 kDa (Fh76335), and ulvan (Yu11689) were purchased from Biosynth. The stock solutions were prepared in PBS at a concentration of 50 mg/ml.

## Mice

C57BL/6J mice were maintained under specific pathogen-free conditions. The animals were housed in ventilated individual cages (IVC system) with controlled temperature (20–24 °C), humidity (45–65%), and a 12:12 h light–dark cycle. The in vivo experiments were performed on female mice aged 8–12 weeks. CD8+ T cells were isolated from C57BL/6J mice or CD45.1+ P14+ mice carrying the transgenic TCR recognizing the LCMV gp33 peptide, and the CD45.1 congenic marker (Pircher et al, 1989). Experiments were performed under the authorization of Landesamt für Natur, Umwelt und Verbraucherschutz Nordrhein-Westfalen in accordance with the German law for animal protection.

## Entry assay

MC57G, BHK, Vero, HEK293T (WT or α-DG ko) cells or BMDCs were seeded into 96-well plates (100,000 cells/well) and treated with different concentrations of tested compounds for 1 h at 4 °C. Next, cells

were incubated with LCMV WE, or rWT and H155Y viruses (MOI 0.5) for 1 h at 4 °C. To initiate infection, cells were moved to 37 °C. At indicated time points, monensin (Thermo Fisher Scientific) was added to block the Golgi apparatus and intracellular protein transport. Cells were then stained with Fixable Viability Dye (Thermo Fisher Scientific), fixed with 2% formaldehyde for 15 min, permeabilised with saponin, and stained with anti-LCMV NP Ab (clone VL4, generated in-house) for 30 min at 4 °C, followed by staining with PE-conjugated anti-rat secondary antibody (Jackson ImmunoResearch, dil. 1:100) according to the standard protocol. The percentage of LCMV NP-positive cells was analyzed using FACS Fortessa and FlowJo software. Representative gating strategies are presented in Appendix Fig. S7.

In an alternative experimental setup, LCMV WE was pretreated with heparin for 1 h at 4 °C, MC57G cells were subsequently incubated with pretreated virus (MOI 0.5) for 1 h at 4 °C, and then the infection was initiated by moving the cells to 37 °C. The following steps were performed as described above.

In the case of the filtration assay, LCMV WE was pretreated with different concentrations of dextran sulfate for 1 h at 4 °C. Next, the samples were filtered through Amicon® Ultra-0.5 Centrifugal Filter Device (100 kDa) three times to remove unbound dextran sulfate. The final solution above the filter was diluted to the initial volume. Next, MC57G cells were incubated with filtered and non-filtered virus samples (MOI 0.5) for 1 h at 4 °C, and then the infection was initiated by moving the cells to 37 °C. The following steps were performed as described above.

## LCMV replication analysis

MC57G cells were seeded into 24-well plates (160,000 cells/well), incubated overnight, and treated with different concentrations of tested GAGs for 1 h. Next, cells were infected with LCMV WE (MOI 0.1) for 3 h. After the infection, cells were washed three times with PBS, and fresh medium containing the tested GAGs was re-added to the cells. After 24 h, supernatants from infected cells were used to determine viral titers using a plaque-forming assay, as described before (Lang et al, 2013). In brief, MC57G cells were added to 24-well plates containing diluted viral samples. After 3 h, a medium with 1% methylcellulose was applied, and plates were further incubated at 37 °C for 48 h. The cells were subsequently fixed with 4% formaldehyde (Sigma-Aldrich) for 30 min, permeabilized with 1% TritonX-100 (Sigma-Aldrich) in HBSS medium (Sigma-Aldrich) for 20 min, and stained with anti-LCMV NP Ab (clone VL4, generated in-house) for 90 min, and peroxidase anti-rat secondary Ab (Jackson ImmunoResearch, dil. 1:300). The viral titers were determined on the basis of plaque formation.

For the heparin treatment, after 24 h, the cells were fixed with 4% formaldehyde, permeabilized with 1% TritonX-100 in HBSS medium, stained with anti-LCMV NP Ab (clone VL4, generated in-house), anti-LCMV GP Ab (clone KL25, generated in-house), and fluorescently-labeled secondary antibodies. Images were acquired with the ZEISS Axio Observer Z1 microscope.

In case of in vivo experiments, organs were extracted into HBSS, homogenized using a Tissue Lyser (QIAGEN), and viral titers were determined by a plaque-forming assay described above.

## RNA purification and RT-PCR

For in vitro experiments with replicating LCMV, MC57G cells were seeded into 24-well plates (160,000 cells/well), incubated overnight,

and treated with different concentrations of tested GAGs for 1 h. Next, cells were infected with LCMV WE (MOI 0.01 or 0.1) for 3 h. After the infection, cells were washed three times with PBS, and fresh medium containing the tested GAGs was re-added to the cells. After 8 and 24 h, the medium was removed, cells were washed three times with PBS, and total cellular RNA was purified using RNeasy Kit (QIAGEN) according to the manufacturer's protocol.

For LCMV entry experiments, MC57G cells were seeded into 24-well plates (160,000 cells/well), incubated overnight, and treated with dextran or dextran sulfate (500 µg/ml) for 1 h. Next, cells were infected with LCMV WE (MOI 0.1) for 5, 15, 45, and 150 min. At indicated time points, the medium was removed, plates were placed on ice for immediate inhibition of infection, and cells were washed 3 times with cold PBS. Total cellular RNA was purified using the RNeasy Kit (QIAGEN) according to the manufacturer's protocol.

In case of in vivo experiments, organs were extracted into lysis buffer from RNeasy Kit (QIAGEN), homogenized using a Tissue Lyser (QIAGEN), and total cellular RNA was purified using RNeasy Kit according to the manufacturer's protocol.

mRNA expression levels were analyzed using iTaq Universal SYBR Green 1-Step Kit (Bio-Rad) according to the manufacturer's protocol. RT-PCR primers were obtained from Eurofins Scientific (Table 1). The expression level of assayed genes was normalized to expression levels of housekeeping genes (*Gapdh* or *Actb*; ΔCt method).

## Elemental analysis

The sulfur content in the tested compounds was determined using a Vario Micro Cube provided by Elementar Analysensysteme GmbH. The measurements were carried out by the Institute for Pharmaceutical and Medicinal Chemistry, Heinrich Heine University Düsseldorf.

## Resazurin assay

To estimate the cytotoxicity of the studied GAGs, the resazurin assay was performed. Cells were seeded into 96-well black plates at a density of $2 \times 10^4$ cells per well, and after 24 h, treated with increasing concentrations of GAGs for 24 or 48 h at 37 °C in an atmosphere of 5% $CO_2$. Following the incubation, resazurin was added to the culture medium to a final concentration of 10 µg/ml, and the plates were incubated at 37 °C to allow conversion of resazurin to resorufin. Fluorescence of metabolized resazurin was measured at two time points at 530 nm excitation and 590 nm emission using Spark Multimode Microplate Reader (Tecan Group, Zürich, Switzerland). Cell viability was calculated as an increase in fluorescence over time, and presented as a percentage of the untreated control.

## Apoptosis assay

MC57G cells were seeded into 96-well plates at a density of $2 \times 10^4$ cells per well, and after 24 h, treated with increasing concentrations of GAGs for 24 h at 37 °C in an atmosphere of 5% $CO_2$. Following the incubation, cells were collected and stained with Annexin V (ImmunoTools) and 7-AAD (Invitrogen) in Annexin V binding Buffer (BD Biosciences) for 20 min at 4 °C. Samples were analyzed using FACS Fortessa and FlowJo software. Representative gating strategies are presented in Appendix Fig. S7.

## Dextran sulfation

Sulfation of dextran was done as previously reported (Soria-Martinez et al, 2020). Briefly, dextran and TMA*$SO_3$ (40 eq. per OH-group) were dissolved in dry dimethylformamide (DMF). The mixture was stirred for 18–19 h at 70 °C. After cooling to room temperature, the mixture was added to 10–20% sodium acetate (10–20 eq. to sulfation agent) at 0 °C. The DMF/water mixture was evaporated under reduced pressure (60 °C, 15 mbar) before dialysis against water (MWCO of 1000–2000 Da) and lyophilization. The degree of sulfation was determined by elemental analysis. NMR spectra confirming the outcome of the reaction are presented in the Appendix Supplementary Information ("Dextran sulfation – elemental analysis and NMR spectrum").

## Synthesis of Man70 and Man70-sulf polymers

The synthesis of the mannose monomer was performed according to Wilkins et al (2015). Details of all further reactions and purity analyses are presented in the Appendix Supplementary Information ("Synthesis of Man70 and Man70-sulf polymers").

## Transmission electron microscopy (TEM)

MC57G cells (700,000/sample) were moved to Eppendorf tubes and treated with dextran or dextran sulfate (500 µg/ml) for 1 h. Next, the cells were infected with LCMV WE (MOI 10) for 1, 5, and 15 min. The cell pellets were fixed using 4% paraformaldehyde, 2.5% glutaraldehyde in 0.1% sodium cacodylate buffer (pH 7.4), and subsequently embedded in a small droplet of 3% low-melting agarose. The pellets were stained with 1% osmium tetroxide for 1 h and with 1% uranyl acetate/1% phosphotungstic acid for 1 h. The samples were dehydrated with a graded ethanol series, embedded in Spurr's resin, and polymerized at 70 °C for 8 h. The ultrathin sections (70 nm) were cut with Ultracut EM UC7 (Leica, Wetzlar, Germany) and transferred to 200-mesh copper grids. These were stained with 1.5% uranyl acetate for 30 min, and with lead citrate for 8 min. The images were acquired using a transmission electron microscope H7100 (Hitachi, Tokyo, Japan) at 100 kV, with a Morada camera (EMSIS GmbH, Münster, Germany) (Lang et al, 2025).

## Arenavirus replication assay

MC57G cells were seeded into 24-well plates (160,000 cells/well), incubated overnight, and treated with heparin, dextran, or dextran sulfate for 1 h. After the incubation, the cells were infected with different Arenaviruses (MOI 0.001). After 1 h, the inoculum was replaced with fresh medium containing the tested GAGs. The experiment was performed in triplicate for each virus and compound concentration. Cell culture supernatant was harvested after 48 h, and titers of infectious virus were measured by immunofocus assay. Briefly, Vero cells were seeded into 96-well plates and inoculated with 50 µl of serial 10-fold dilutions of the supernatant sample. The inoculum was removed after 1 h and

replaced with overlay medium (DMEM with 10% FCS) containing 1% methylcellulose. After 3 days of incubation, cells were fixed with 4% formaldehyde, washed with PBS, and permeabilized with 0.5% TritonX-100 in PBS. After washing with PBS and blocking with 2% FCS in PBS, cell foci were detected with NP-specific monoclonal antibodies (clone 2B5, generated in-house for LASV and MORV; clone 2LD9, generated in-house for LUJV) or GP-specific monoclonal antibodies (clone IC06-BE10, obtained from BEI Resources, NIAID, NIH, for MACV, JUNV, TACV, and PARV). After washing, cells were incubated with peroxidase-labeled anti-mouse IgG. Foci were visualized with tetramethylbenzidine and counted.

## GP-Fc binding assay

MC57G cells were seeded into 24-well plates (150,000 cells/well), and after 24 h, pretreated with dextran and dextran sulfate. Cells were subsequently treated with recombinant LCMV glycoprotein supernatant derived from HEK293 cells expressing LCMV GP1-Fc (prepared in-house; Recher et al, 2004; Khairnar et al, 2015) (at 4 °C to prevent internalization). Then the cells were washed twice with fresh medium to remove unbound GP1-Fc, and stained with PE-conjugated goat anti-human anti-Fc antibody (Invitrogen). Images were acquired with the ZEISS Axio Observer Z1 microscope and analyzed with ImageJ software.

## Structure modeling

To investigate the interaction of sulfated and non-sulfated dextran polymers and LCMV GP, an initial model of the extracellular domain of the LCMV GP trimer was taken from the cryo-EM structure (PDB ID: 8DMI) (Moon-Walker et al, 2023), with missing residues added using MODELLER (Šali and Blundell, 1993). Additionally, N-glycan chains (Appendix Fig. S4) were attached to the 9 glycosylation sites on each of the protomers using the tLEaP module as part of the AMBER molecular dynamics suite (Woods, 2005; Case et al, 2023). The respective glycosylation patterns were adopted from those described previously for LASV GP (Re and Mizuguchi, 2021).

A model of the sulfated dextran polymer with a molecular weight of 5 kDa (corresponding to a 12mer) and a non-sulfated dextran polymer consisting of an equal amount of glucose units was generated using the GLYCAM web interface (Woods, 2005). Both polymers were modeled as linear chains of α-1,6-linked glucose units.

In preparation for performing molecular dynamics (MD) simulations, a single (sulfated) dextran polymer was randomly positioned at least 40 Å from the LCMV GP surface using PACKMOL (Martínez et al, 2009). This way, ten different starting points were created for either dextran polymer (Appendix Fig. S5). Each system was solvated using tLEaP, thereby ensuring a distance of at least 10 Å between the dextran chain and the edge of the solvent box. Finally, solvent molecules were replaced by counter ions (KCl 0.15 M) to obtain a neutral system, yielding systems of ~650,000 atoms.

## Molecular dynamics simulations

Unbiased molecular dynamics simulations of the LCMV GP/ dextran systems were performed using the GPU implementation of PMEMD (Salomon-Ferrer et al, 2013) as part of version 23.4 of the AMBER molecular dynamics suite (Case et al, 2023), using the GLYCAM06 (Kirschner et al, 2008) and ff19SB (Tian et al, 2019) force field for the polysaccharide chains and protein, respectively. Water molecules and ions were parametrized using the TIP3P model (Zhao et al, 2019) and Li and Merz 12-6 ions parameters (Li et al, 2015). Simulations were carried out at a target temperature of 300 K maintained using the Langevin thermostat (Loncharich et al, 1992) with a collision frequency of $1.0 \text{ ps}^{-1}$. Covalent bonds to hydrogen atoms were constrained using the SHAKE algorithm (Ryckaert et al, 1977) with a tolerance of $10^{-5}$ Å. Topology files with repartitioned hydrogen masses were used to allow a timestep of 4 fs.

Each system was subjected to an initial minimization of solvent molecules for 2500 cycles using the steepest descent algorithm, followed by 2500 minimization cycles using the conjugate gradient algorithm. The movement of polysaccharide chains and protein residues was restrained by applying harmonic potentials with force constants ranging from 2.5 to 10 kcal mol$^{-1}$ Å$^{-2}$ (Appendix Table S1). The system was then thermalized from 100 to 300 K for 50.0 ps under NVT conditions with identical restraints, and a timestep of 1 fs was applied. Subsequently, a total of seven equilibration steps (total simulation time: 950 ps) were performed in the NPT ensemble, gradually lowering the restraints and increasing the simulation timestep to 4 fs by using hydrogen mass repartitioning. The equilibrated systems were then subjected to production runs of 1.5 μs length each.

Post-processing and analysis of MD trajectories were carried out using CPPTRAJ (Roe and Cheatham, 2013) as part of version 23.4 of the AMBER molecular dynamics suite. Visual inspection of trajectories was performed using VMD (Humphrey et al, 1996) and PyMOL (Schrodinger, 2015).

## IFN-α ELISA

IFN-α levels in serum samples were analyzed with the Mouse IFN alpha ELISA Kit (Thermo Fisher Scientific) according to the manufacturer's protocol.

## Histology

Snap-frozen tissue sections were fixed in 100% acetone at room temperature (RT) for 30 min. Following fixation, the cryosections were washed three times for 5 min in PBS, blocked with 10% FCS in PBS (with Fc-block) for 40 min at RT, stained with primary antibodies conjugated with fluorophores at RT for 1 h, followed by three washes with PBS. All antibodies were diluted in 1% FCS in PBS, 1:50 or 1:100. Sections were mounted with Vectashield Vibrance® Antifade Mounting Medium (Vector Labs, US) with DAPI. Images were acquired with the ZEISS Axio Observer Z1 microscope and analyzed with ImageJ software. JACoP plugin in ImageJ was used to determine co-localization correlation from the images by calculating Pearson's correlation coefficient.

## Liver damage analysis

The serum activities of liver transaminases (aspartate transaminase (AST) and alanine transaminase (ALT)) were determined using the SpotChem™ EZ Clinical Chemistry Analyser with dedicated liver profile kits (Arkray) according to the manufacturer's protocol.

## Flow cytometry

Tetramer and intracellular cytokine staining were performed as described previously (Xu et al, 2024). For tetramer staining, single-cell suspensions were incubated with tetramer-gp33 and -np396 for 15 min at 37 °C. After incubation, antibodies conjugated with fluorophores (anti-CD8, CD19, 2B4, IL7R, KLRG1, and TIM-3 from Invitrogen; CD44, CD62L, PD-1, and LAG3 from BD Biosciences) were added for 30 min at 4 °C. For intracellular cytokine staining, single-cell suspensions were stimulated with LCMV-specific peptides (gp33-41 and np396-404) for 1 h. Brefeldin A (Thermo Fisher Scientific) was added for another 5 h incubation at 37 °C, followed by overnight staining with anti-CD8 antibody (Thermo Fisher Scientific), fixation with 2% formaldehyde, permeabilisation with saponin, and staining with anti-IFN-γ and anti-TNF-α antibodies (Thermo Fisher Scientific). Samples were analyzed using FACS Fortessa and FlowJo software. Representative gating strategies are presented in Appendix Fig. S7. The expression of CD44, CD62L, IL7R and KLRG1 has been additionally reported as the frequency of tet+ cells and MFI in Appendix Figs. S8 and S9.

## Purification of T cells

Following the manufacturer's recommendations, single-cell suspended splenocytes and lymph node cells were enriched with the mouse CD8 purification kit (Miltenyi Biotec).

## T-cell proliferation and differentiation analysis

CD8+ T cells were isolated from C57BL/6J mice, activated with anti-CD3/CD28 (2 μg/1 μg) antibodies (Invitrogen/BD Biosciences), and at the same time treated with dextran or dextran sulfate. For proliferation analysis, the cells were additionally stained with cell proliferation dye (Invitrogen) for 10 min at 37 °C. After 48 h, the cells were incubated with anti-CD8, CD62L, and CD44 antibodies for 30 min at 4 °C. Proliferation and T-cell populations were analyzed using FACS Fortessa and FlowJo software. Representative gating strategies are presented in Appendix Fig. S7.

## Generation of bone marrow-derived dendritic cells (BMDCs)

BMDCs were prepared as described before (Lutz et al, 1999; Xu et al, 2021). Bone marrow cells from C57BL/6J mice were flushed from femurs and tibias. The cells were seeded at a concentration of $2 \times 10^6$ cells/10 ml in tissue-culture plates and differentiated with GM-CSF (PAN-Biotech, 40 ng/ml) in RPMI1640 medium containing 10% fetal bovine serum (Gibco), 1% penicillin-streptomycin-glutamine (Thermo Fisher Scientific), and 1 mM β-mercaptoethanol (Sigma-Aldrich), with a medium change on day 3, 6, and 8. BMDCs were harvested on day 10 for experiments.

## BMDC activation and antigen presentation assay

BMDCs were treated with dextran and dextran sulfate for 1 h at 37 °C. Cells were subsequently infected with LCMV WE (MOI 0.5) for 1 h at 37 °C. Then the dextrans and virus were removed by washing. CD8+ P14+ T cells were isolated and mixed with BMDCs at a ratio of 10:1. After 72 h of incubation, Brefeldin A (Thermo Fisher Scientific) was added for another 5 h incubation at 37 °C,

followed by overnight staining with anti-CD8 antibody (Thermo Fisher Scientific), fixation with 2% formaldehyde, permeabilisation with saponin, and staining with anti-IFN-γ and anti-Ki67 antibodies (Thermo Fisher Scientific). Samples were analyzed using FACS Fortessa and FlowJo software. Representative gating strategies are presented in Appendix Fig. S7.

In an analogous setting, CD8+ P14+ T cells were stained with cell proliferation dye (10 min, 37 °C) and mixed with BMDCs at a ratio of 10:1. After 72 h of incubation, T cells were incubated with anti-CD8, CD62L, and CD44 antibodies for 30 min at 4 °C, and proliferation and T-cell populations were analyzed using FACS Fortessa and FlowJo software. Representative gating strategies are presented in Appendix Fig. S7.

In the setting without the co-culture with T cells, surface expression of MHC-I, MHC-II, and CD86 was analyzed on CD11c+ cells after treatment with dextran/dextran sulfate, infection with LCMV WE (MOI 0.5), washing, and culturing in fresh medium for 24 h.

## Statistical analysis and experimental design

No formal randomization was applied, as the study did not involve randomized clinical interventions. Investigators were not blinded to group allocation during data acquisition or analysis. Experiments

### The paper explained

#### Problem

Although rodent species are the primary host for Arenaviruses, some strains may cause severe and potentially life-threatening diseases in humans. Despite promising preclinical studies, few vaccine candidates have reached human clinical trials, particularly for Lassa fever, although the success of Candid #1 against Argentine hemorrhagic fever-causing Junín virus provides hope that similar approaches could work for other Arenavirus species.

Arenaviruses initiate infection by binding to specific cellular receptors on host cells, such as α-dystroglycan, but can also exploit cell-surface proteoglycans and glycosaminoglycans (GAGs) to facilitate viral attachment. Understanding how these interactions contribute to viral entry could reveal new targets for therapeutic intervention.

#### Results

This study identifies highly sulfated GAGs as potent inhibitors of Arenavirus cell entry. Among the tested compounds, dextran sulfate showed the strongest antiviral activity, interfering with the binding of the viral glycoprotein to its cellular receptors. This prevented virus-receptor engagement and efficiently blocked infection in vitro for both pathogenic and non-pathogenic Arenavirus species. In infected mice, dextran sulfate treatment during the active phase of infection significantly reduced viral titers in the spleen and liver, alleviated tissue pathology, and enhanced antiviral T-cell responses. By contrast, early administration before immune activation limited antigen presentation, resulting in reduced T-cell priming and prolonged viral persistence, underscoring the importance of treatment timing.

#### Impact

These findings reveal a dual role of sulfated glycans in Arenavirus infection – both as inhibitors of viral entry and as modulators of antiviral immunity. By uncovering the molecular mechanism of GAG-mediated viral blockade and demonstrating therapeutic efficacy in vivo, this study provides a foundation for developing sugar-based antiviral agents. Such compounds could offer a novel, broadly applicable strategy against Arenavirus infections and potentially other viruses that rely on charge-dependent receptor interactions.

were independently replicated as indicated for each assay, and the exact number of replicates (*n*) is specified in the corresponding figure legends. Statistical analyses were performed using appropriate tests as specified in the figure legends. Exact *P* values are provided in the Appendix (Appendix Table S2). All additional information, including details on animal models, cell lines, viruses, reagents, and experimental procedures, is provided in the relevant sections of the "Methods" and the Table of Reagents.

## Graphics

Schematic representations of experimental setups and synopsis figures were created, in part, using BioRender.com.

## Data availability

The modeling data produced in this study are available in the following database: https://doi.org/10.25838/d5p-82.

The source data of this paper are collected in the following database record: biostudies:S-SCDT-10_1038-S44321-026-00387-8.

## Peer review information

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

## Acknowledgements

This study was supported by the German Research Foundation (DFG, GRK1949, LA2558/8-1), the Jürgen Manchot Foundation (Molecules of Infection), the Volkswagen Foundation (9B797), the Medical Faculty of Heinrich Heine University (Forschungskommission, project 9772816), and the NIH Tetramer Core Facility. PAL was further supported by the Christiane and Claudia Hempel Foundation and the BMBF (14LW0450). This study was further funded by the German Research Foundation (DFG) under the DFG Priority Research Program SPP 2416 – CodeChi – project ID 525826480 (GO 1367/7-1 to HG) and a grant from the Ministry of Innovation, Science, and Research of North-Rhine Westphalia (NRW) within the framework of the NRW Strategieprojekt BioSC (No. 313/323-400-002 13) by the BOOST FUND 2.0 project "OptiCellu" (to HG). UGD and LH thank the SFF – Strategic Research Fund of Heinrich Heine University Düsseldorf for financial support. The authors are grateful for the computational infrastructure and support provided by the "Zentrum für Informations- und Medientechnologie" (ZIM) at Heinrich Heine University Düsseldorf and the computing time provided by the John von Neumann Institute for Computing (NIC) to HG on the supercomputer JUWELS at Jülich Supercomputing Centre (JSC) (user IDs: lignin, chitin). We acknowledge the Core Facility Electron Microscopy (CFEM) at the Medical Faculty of Heinrich Heine University Düsseldorf for support with EM imaging. The following reagent was obtained through BEI Resources, NIAID, NIH: Monoclonal Anti-Junin Virus, clone IC06-BE10 (produced in vitro), NR-48834. We acknowledge technical support from Filip Koenig (HHU Düsseldorf).

## Author contributions

Michal Gorzkiewicz: Conceptualization; Resources; Data curation; Formal analysis; Supervision; Funding acquisition; Validation; Investigation; Visualization; Methodology; Writing—original draft; Writing—review and editing; Project administration. Soha Noseir: Conceptualization; Data curation; Investigation; Methodology; Project administration; Writing—review and editing. Mandar Vengurlekar: Data curation; Formal analysis; Validation; Investigation; Visualization; Methodology; Writing—review and editing. Mitrajit Ghosh: Data curation; Formal analysis; Validation; Investigation; Visualization; Methodology; Writing—review and editing. Ichiro Katahira: Data curation; Investigation. Džiuljeta Abromavičiūtė: Data curation; Investigation. Ulla Gerling-Driessen: Data curation; Formal analysis; Validation; Investigation; Visualization; Methodology; Writing—original draft; Writing—review and editing. Lorand Bonda: Data curation; Formal analysis; Investigation; Writing—review and editing. Nick Rähse: Data curation; Formal analysis; Validation; Investigation; Visualization; Writing—original draft; Writing—review and editing. Marco Lapsien: Data curation; Formal analysis; Validation; Investigation; Visualization; Writing—original draft; Writing—review and editing. Sabrina Bockholt: Data curation; Investigation. Ann Kathrin Bergmann: Resources; Data curation; Software; Validation; Investigation; Visualization; Methodology. Konstantina Kostadinovska: Data curation; Investigation. Hafssa Fraii: Data curation; Investigation. Karl S Lang: Resources; Writing—review and editing. Lisa Oestereich: Resources; Data curation; Formal analysis; Supervision; Validation; Investigation; Visualization; Methodology; Writing—review and editing. Holger Gohlke: Resources; Data curation; Software; Formal analysis; Supervision; Funding acquisition; Validation; Investigation; Visualization; Methodology; Writing—original draft; Writing—review and editing. Laura Hartmann: Resources; Data curation; Formal analysis; Supervision; Validation; Investigation; Methodology; Writing—review and editing. Philipp A Lang: Conceptualization; Resources; Software; Formal analysis; Supervision; Funding acquisition; Validation; Investigation; Methodology; Writing—original draft; Project administration; Writing—review and editing.

Source data underlying figure panels in this paper may have individual authorship assigned. Where available, figure panel/source data authorship is listed in the following database record: biostudies:S-SCDT-10_1038-S44321-026-00387-8.

## Funding

## Disclosure and competing interests statement

KSL and PAL declare a financial conflict of interest as cooperation partners of, advisors to, founders, shareholders, and patent inventors of Abalos Therapeutics GmbH, developing an Arenavirus-based cancer therapy. The remaining authors declare no competing interests.

# Expanded View Figures

**Figure EV1.  Inhibition of Arenavirus infection by GAGs depends on their structure and sulfation.**

(**A**) Chemical structures of GAGs used in the study. (**B, C**) MC57G cells were treated with dextran and dextran sulfate (500 µg/ml) for 1 h. Next, cells were infected with LCMV WE (MOI 0.01) for 3 h. After the infection, cells were washed three times with PBS and fresh medium containing studied compounds was re-added to the cells. The cells were collected after 8 (**B**) and 24 h (**C**) for RNA isolation and RT-PCR. Data presented as number of cognate mRNA copies per copy of mRNA for reference housekeeping gene, mean ± SEM, $n = 3$. *$P < 0.05$, **$P < 0.01$ compared to samples treated with dextran. Statistical significance was assessed by Student's *t* test. (**D, E**) MC57G cells were treated with dextran and dextran sulfate (500 µg/ml) for 1 h. Next, cells were infected with LCMV WE (MOI 0.1) for 3 h. After the infection, cells were washed three times with PBS and fresh medium containing studied compounds was re-added to the cells. The cells were collected after 8 (**D**) and 24 h (**E**) for RNA isolation and RT-PCR. Data presented as number of cognate mRNA copies per copy of mRNA for reference housekeeping gene, mean ± SEM, $n = 3$. *$P < 0.05$, **$P < 0.01$, ***$P < 0.001$ compared to samples treated with dextran. Statistical significance was assessed by Student's *t* test. (**F**) BHK cells were treated with Man70 or Man70-sulf for 1 h. Next, cells were infected with LCMV WE (MOI 0.5) for 1 h at 4 °C. After the infection, cells were incubated at 37 °C for 2.5 h before adding monensin to block the Golgi apparatus and intracellular protein transport. Cells were then stained for viability and LCMV NP. Data presented as mean ± SEM, $n = 9$–12. **$P < 0.01$, ***$P < 0.001$ compared to vehicle control. Statistical significance was assessed by one-way ANOVA. (**G**) MC57G cells were treated with heparin, dextran, or dextran sulfate and infected with Tacaribe (TCRV), Parana (PARV), and Morogoro (MORV) viruses (BSL2) at MOI of 0.001. Cell culture supernatant was harvested after two days and virus titers were determined by immunofocus assay. Data presented as focus-forming units (ffu) per ml, mean ± SEM $n = 3$. *$P < 0.05$, **$P < 0.01$, ***$P < 0.01$ compared to dextran treatment. Statistical significance was assessed by two-way ANOVA. Source data are available online for this figure.

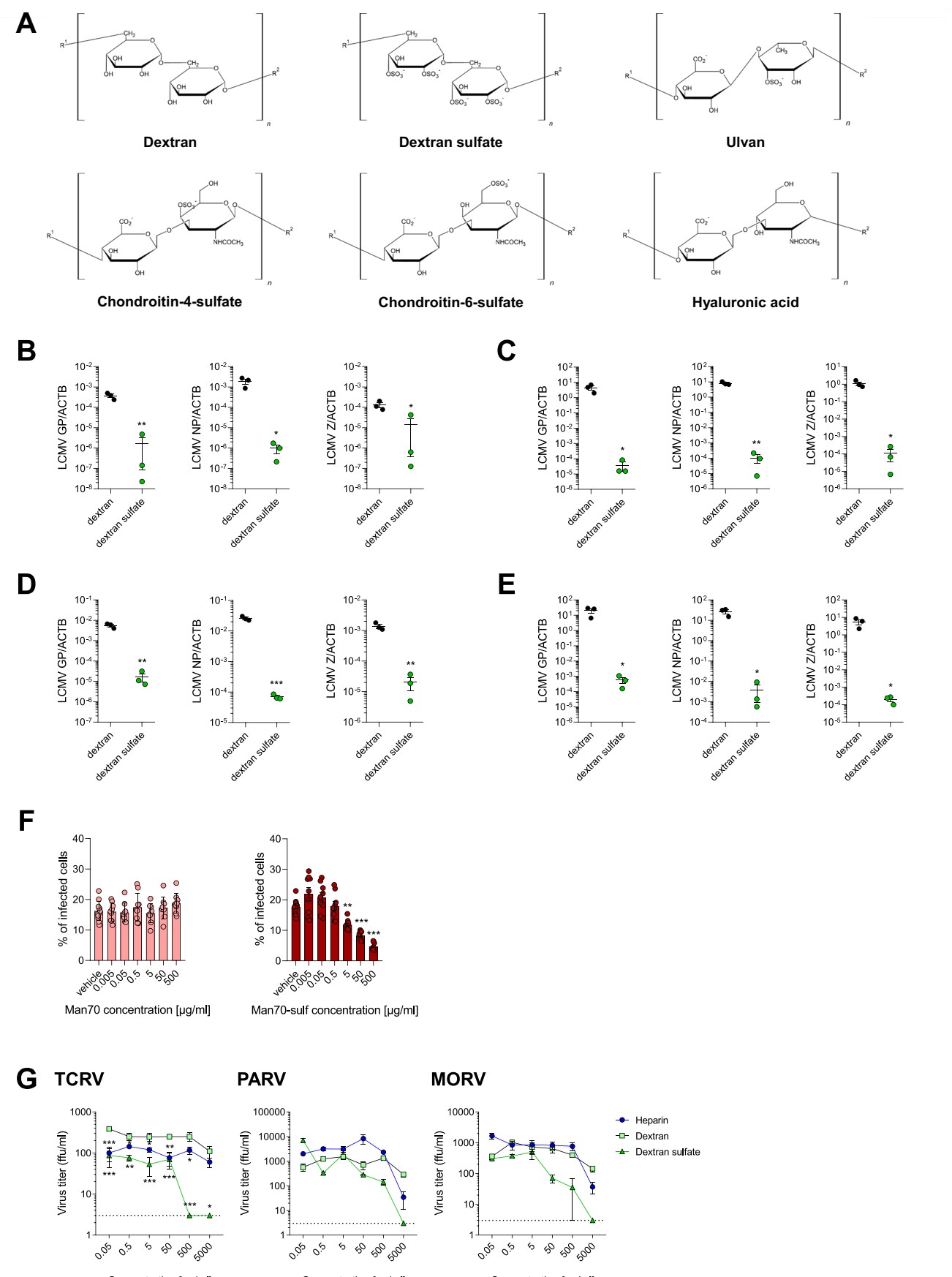

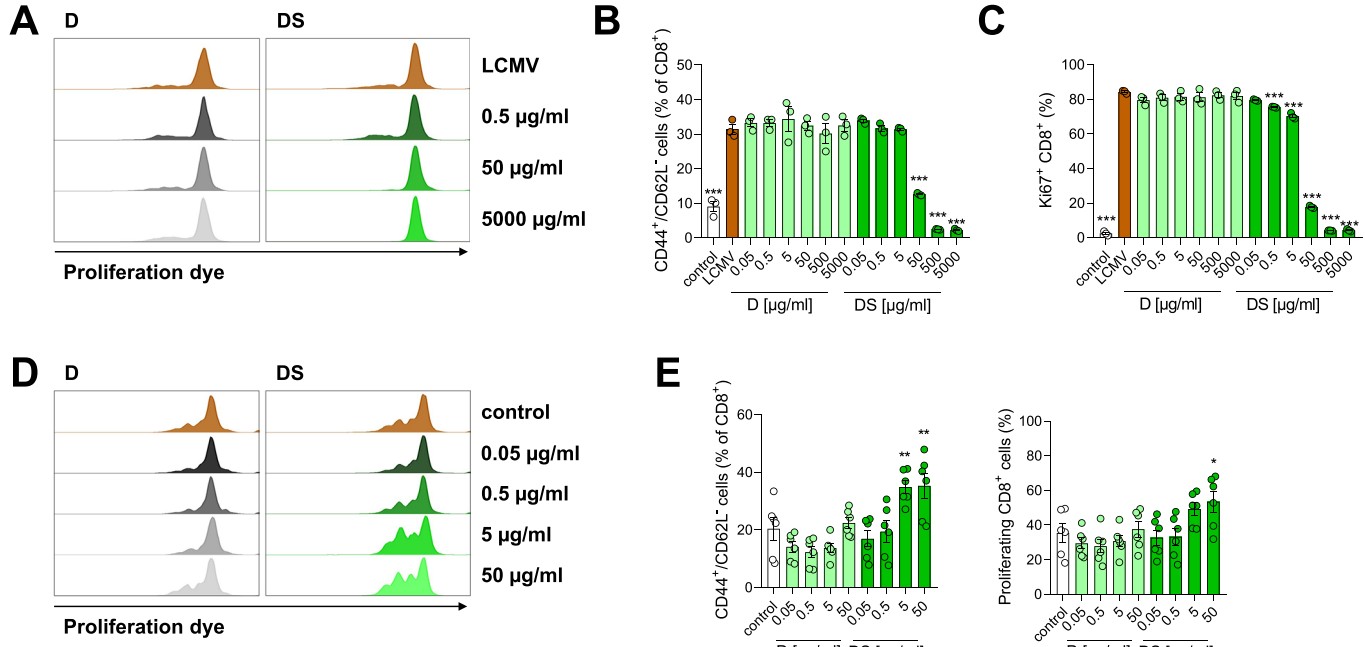

**Figure EV2.  Dextran sulfate decreases infectivity in BMDCs causing impaired antigen presentation and T-cell function.**

(A, B) BMDCs were treated with dextran/dextran sulfate for 1 h. Cells were subsequently infected with LCMV WE (MOI 0.5) for 1 h. Then the dextrans and virus were removed by washing. CD8⁺ P14⁺ T cells were isolated, stained with cell proliferation dye (10 min, 37 °C) and mixed with BMDCs at a ratio of 10:1. After 72 h of incubation, T cells were analyzed for proliferation (A) and effector populations (B). For proliferation, representative histograms are shown ($n = 3$). For effector populations, data presented as mean ± SEM, $n = 3$. ***$P < 0.001$ compared to LCMV-infected control. Statistical significance was assessed by one-way ANOVA. (C) BMDCs were treated with dextran/dextran sulfate for 1 h. Cells were subsequently infected with LCMV WE (MOI 0.5) for 1 h. Then the dextrans and virus were removed by washing. CD8⁺ P14⁺ T cells were isolated and mixed with BMDCs at a ratio of 10:1. After 72 h of incubation, T cells were analyzed for Ki67 expression. Data presented as mean ± SEM, $n = 3$. ***$P < 0.001$ compared to LCMV-infected control. Statistical significance was assessed by one-way ANOVA. (D, E) CD8⁺ T cells were isolated from spleen and lymph nodes of C57BL/6J mice, activated with anti-CD3/CD28 antibodies, and at the same time treated with dextran (D) or dextran sulfate (DS). For proliferation analysis, the cells were additionally stained with cell proliferation dye for 10 min at 37 °C. After 48 h, the cells were incubated with anti-CD8, CD62L, CD44 antibodies for 30 min at 4 °C. Proliferation and T-cell populations were analyzed by flow cytometry. (D) Representative proliferation histograms are shown. (E) Data presented as mean ± SEM, $n = 6$. *$P < 0.05$, **$P < 0.01$, compared to non-treated control. Statistical significance was assessed by one-way ANOVA. Source data are available online for this figure.

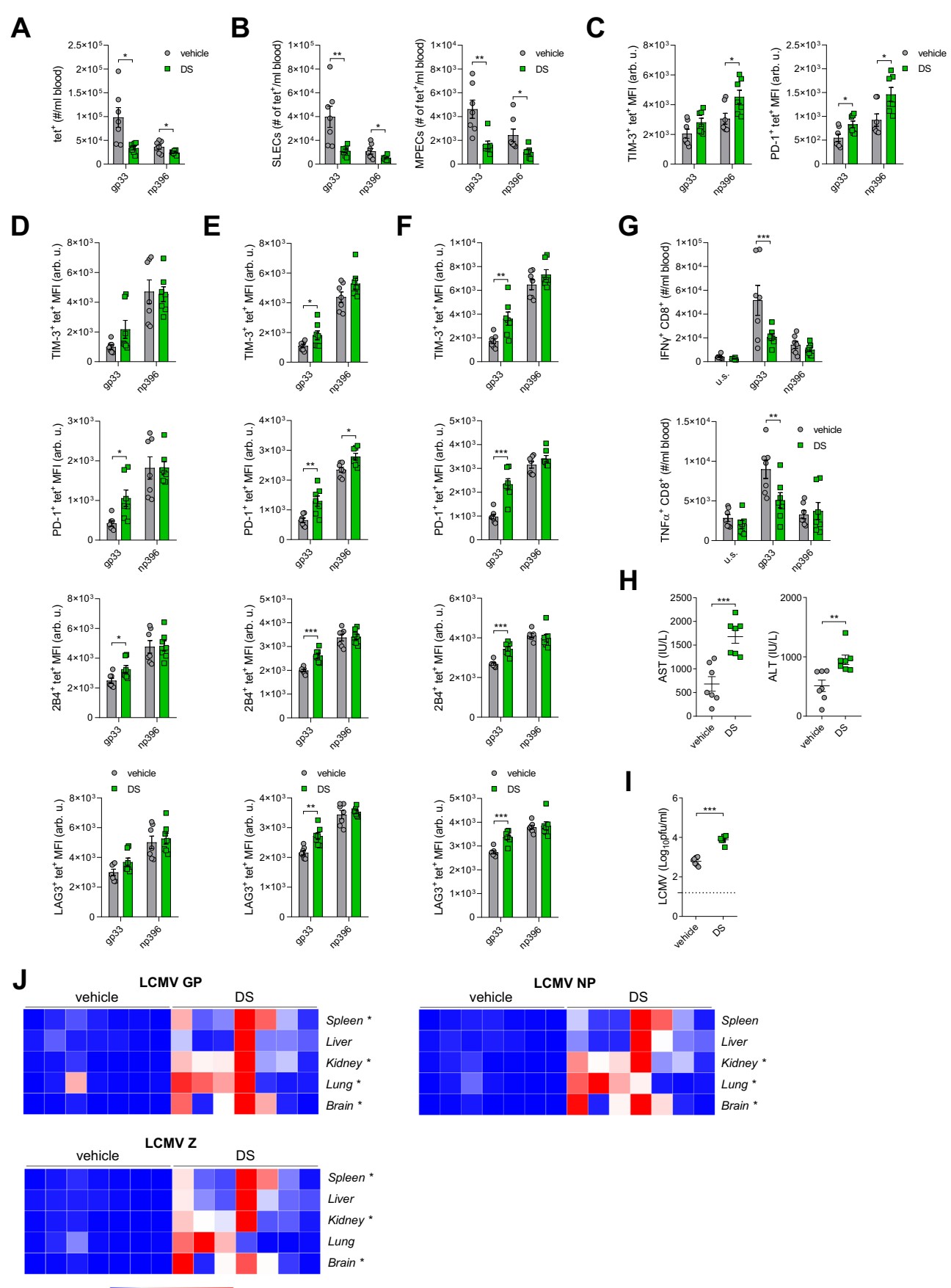

◀ **Figure EV3. Treatment with dextran sulfate at the beginning of infection leads to virus persistence and pathology.**

LCMV WE ($5 \times 10^6$ pfu) was pretreated with dextran sulfate (DS, 500 µg/ml) for 30 min, and then injected i.v. into C57BL/6J mice ($10^6$ pfu per mouse), while control mice were infected with LCMV WE. Mice were re-injected i.v. with dextran sulfate (100 µg per mouse) 6 h p.i., and control mice were injected with PBS. Analyses were carried out on day 8 and 12 p.i. (A) Tet-gp33$^+$ and tet-np396$^+$ T cells were determined in blood on day 8 p.i. (B) SLECs (IL7R$^-$, KLRG1$^+$) and MPECs (IL7R$^+$, KLRG1$^-$) subsets within tet$^+$ T cells were determined in blood on day 8 p.i. (C) Surface expression of exhaustion markers on tet$^+$ T cells was determined in blood on day 8 p.i. (D–F) Surface expression of exhaustion markers on tet$^+$ T cells was determined in blood (D), spleen (E), and liver (F) on day 12 p.i. (G) IFN-γ and TNF-α production by CD8$^+$ T cells in blood after re-stimulation with LCMV-specific peptides was determined on day 8 p.i. (H) ALT and AST activity in the serum of control and DS-treated mice was evaluated on day 8 p.i. (I) LCMV titer was determined in blood on day 8 p.i. Tet$^+$ CD8$^+$ T cells, SLECs, MPECs (both populations as subsets of tet$^+$ cells), and cytokine-producing CD8$^+$ T cells are presented as absolute counts: a number of cells per ml of blood, per spleen, or per liver lobe, as indicated. Data presented as mean ± SEM, $n = 7$ mice per condition, *$P < 0.05$, **$P < 0.01$, ***$P < 0.001$. Statistical significance was assessed by Student's *t* test, or two-way ANOVA in case of IFN-γ and TNF-α production. (J) mRNA expression of LCMV genes was determined in spleen, liver, kidney, lung and brain tissue. Data presented as heatmaps representing the number of cognate mRNA copies per copy of mRNA for reference housekeeping gene, $n = 7$ mice per condition. *$P < 0.05$ between control and DS-treated groups. Statistical significance was assessed by Student's *t* test. Source data are available online for this figure.

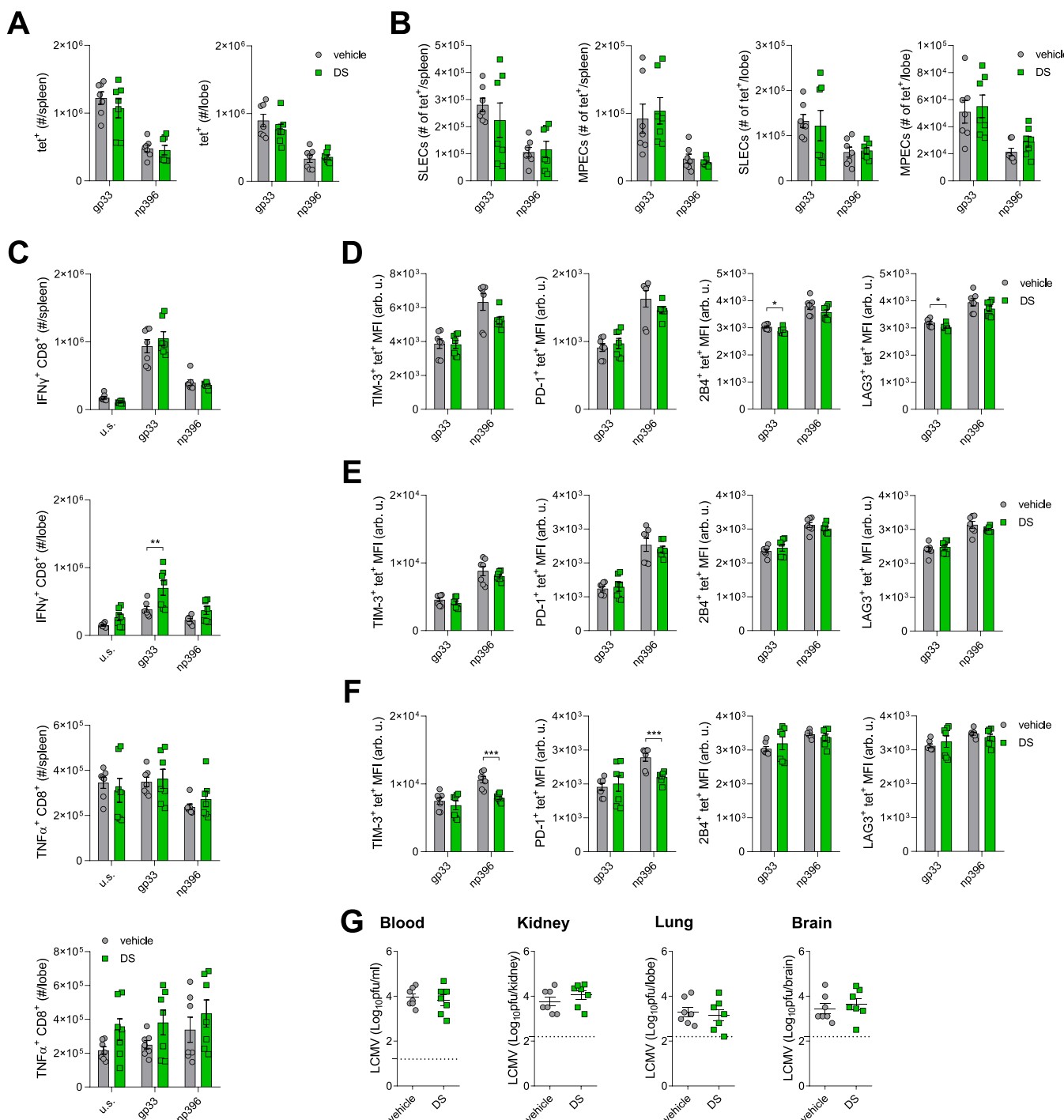

**Figure EV4. Treatment with dextran sulfate in the course of active infection leads to increased antiviral immunity.**

C57BL/6J mice were infected i.v. with LCMV WE ($10^6$ pfu per mouse). On day 6 and 7 p.i., mice were injected i.v. with dextran sulfate (DS, 100 μg per mouse), while control mice were injected with PBS. Analyses were carried out on day 8 p.i. (A) Tet-gp33+ and tet-np396+ T cells were determined in spleen and liver. (B) SLECs (IL7R-, KLRG1+) and MPECs (IL7R+, KLRG1-) subsets within tet+ T cells were determined in spleen and liver. (C) IFN-γ and TNF-α production by CD8+ T cells in spleen and liver after re-stimulation with LCMV-specific peptides was determined. (D–F) Surface expression of exhaustion markers on tet+ T cells was determined in blood (D), spleen (E), and liver (F). (G) LCMV titers were determined in blood, kidney, lung and brain. Tet+ CD8+ T cells, SLECs, MPECs (both populations as subsets of tet+ cells), and cytokine-producing CD8+ T cells are presented as absolute counts: number of cells per ml of blood, per spleen, or per liver lobe, as indicated. Data presented as mean ± SEM, $n = 7$ mice per condition, *$P < 0.05$, **$P < 0.01$, ***$P < 0.001$. Statistical significance was assessed by Student's $t$ test, or two-way ANOVA in case of IFN-γ and TNF-α production. Source data are available online for this figure.

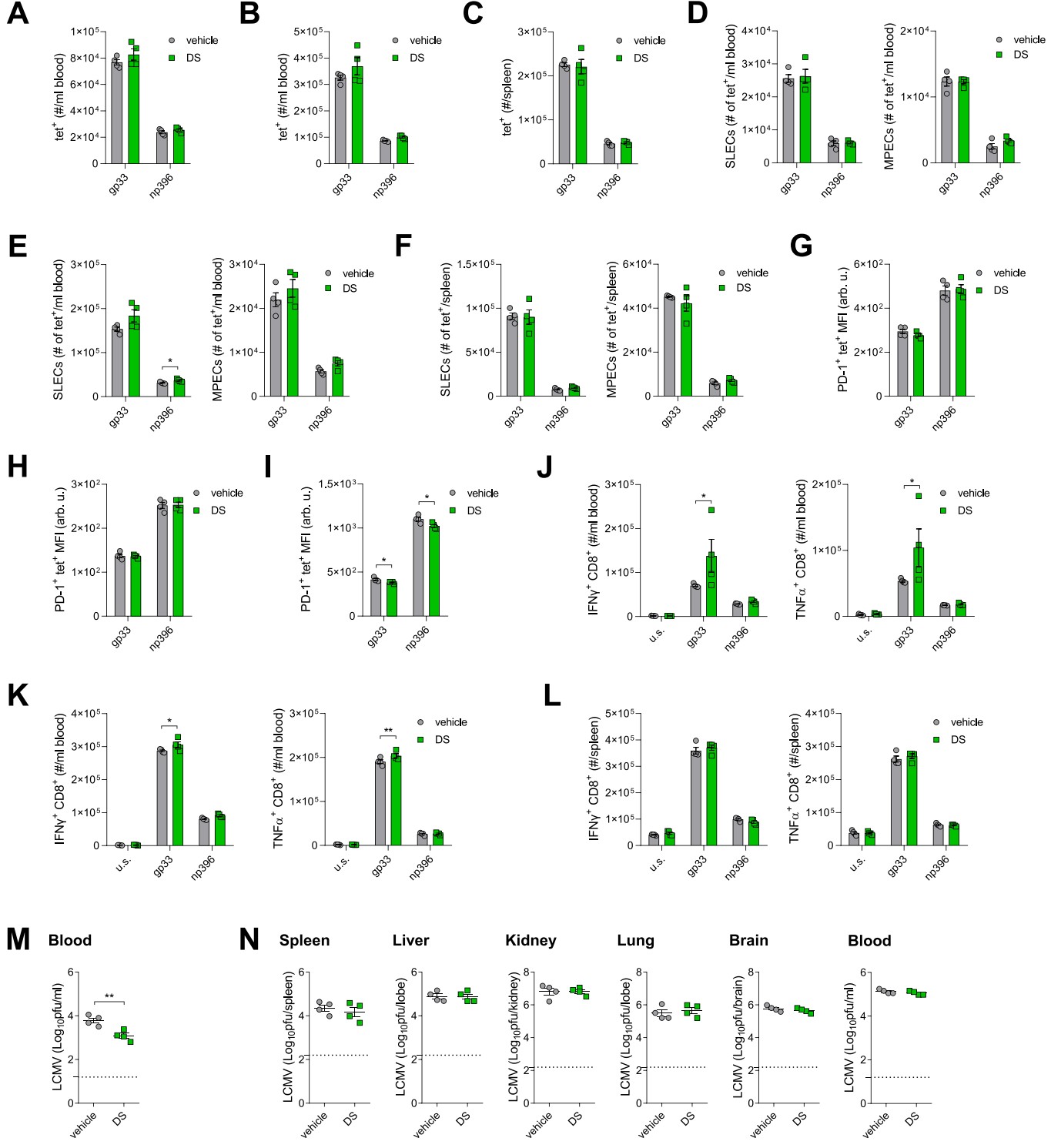

**Figure EV5. Treatment with dextran sulfate in the course of LCMV clone 13 infection leads to transiently increased antiviral immunity.**

C57BL/6J mice were infected i.v. with LCMV clone 13 ($10^6$ pfu per mouse). On day 6 and 7 p.i., mice were injected i.v. with dextran sulfate (DS, 100 μg per mouse), while control mice were injected with PBS. Analyses were carried out on day 8, 12 and 20 p.i. (**A–C**) Tet-gp33$^+$ and tet-np396$^+$ T cells were determined in blood on day 8 (**A**), day 12 (**B**), and in spleen (**C**) on day 20 p.i. (**D–F**) SLECs (IL7R$^-$, KLRG1$^+$) and MPECs (IL7R$^+$, KLRG1$^-$) subsets within tet$^+$ T cells were determined in blood on day 8 (**D**), day 12 (**E**), and in spleen (**F**) on day 20 p.i. (**G–I**) Surface expression of PD-1 on tet$^+$ T cells was determined in blood on day 8 (**G**), day 12 (**H**), and in spleen (**I**) on day 20 p.i. (**J–L**) IFN-γ and TNF-α production by CD8$^+$ T cells in blood on day 8 (**J**), day 12 (**K**), and in spleen (**L**) on day 20 p.i. after re-stimulation with LCMV-specific peptides was determined. (**M**) LCMV titers were determined in blood on day 8 p.i. (**N**) LCMV titers were determined in blood, spleen, liver, kidney, lung and brain on day 20 p.i. Tet$^+$ CD8$^+$ T cells, SLECs, MPECs (both populations as subsets of tet$^+$ cells), and cytokine-producing CD8$^+$ T cells are presented as absolute counts: number of cells per ml of blood, or per spleen, as indicated. Data presented as mean ± SEM, $n = 4$ mice per condition, *$P < 0.05$, **$P < 0.01$. Statistical significance was assessed by Student's $t$ test, or two-way ANOVA in case of IFN-γ and TNF-α production. Source data are available online for this figure.

