## [Peer Review File · EMBO Molecular Medicine]

Sulfated glycosaminoglycans inhibit LCMV entry and modulate antiviral immunity and pathology

Michal Gorzkiewicz, Soha Noseir, Mandar Vengurlekar, Mitrajit Ghosh, Ichiro Katahira, Džuljeta Abromavičiūtė, Ulla Gerling-Driessen, Lorand Bonda, Nick Rähse, Marco Lapsien, Sabrina Bockholt, Ann Kathrin Bergmann, Konstantina Kostadinovska, Hafssa Fraii, Karl Lang, Lisa Oestereich, Holger Gohlke, Laura Hartmann, and Philipp Lang

Corresponding author: Philipp Lang (langp@uni-duesseldorf.de)

Review Timeline:

Submission Date:	20th May 25
Editorial Decision:	1st Jul 25
Revision Received:	21st Oct 25
Editorial Decision:	8th Dec 25
Revision Received:	25th Jan 26
Accepted:	27th Jan 26

Editor: Zeljko Durdevic

Transaction Report:

1st Jul 2025

Dear Prof. Lang,

Thank you for the submission of your manuscript to EMBO Molecular Medicine. We have now received feedback from the three reviewers who agreed to evaluate your manuscript. As you will see from the reports, all three referees recognize potential interest of the study, but they also raise serious and partially overlapping concerns that should be addressed in a major revision. If you would like to discuss further the points raised by the referees, I am available to do so via email or video. Let me know if you are interested in this option.

We would welcome the submission of a revised version within three months for further consideration. Please let us know if you require longer to complete the revision.

I look forward to receiving your revised manuscript.

Yours sincerely,

Zeljko Durdevic

Zeljko Durdevic
Senior Editor
EMBO Molecular Medicine

We require:

- 1) A .docx formatted version of the manuscript text (including legends for main figures, EV figures and tables). Please make sure that the changes are highlighted to be clearly visible.
- 2) Individual production quality figure files as .eps, .tif, .jpg (one file per figure). For guidance, download the 'Figure Guide PDF': (<https://www.embopress.org/page/journal/17574684/authorguide#figureformat>).
- 3) A .docx formatted letter INCLUDING the reviewers' reports and your detailed point-by-point responses to their comments. As part of the EMBO Press transparent editorial process, the point-by-point response is part of the Review Process File (RPF), which will be published alongside your paper.
- 4) A complete author checklist, which you can download from our author guidelines (<https://www.embopress.org/page/journal/17574684/authorguide#submissionofrevisions>). Please insert information in the checklist that is also reflected in the manuscript. The completed author checklist will also be part of the RPF.
- 5) Please note that all corresponding authors are required to supply an ORCID ID for their name upon submission of a revised manuscript.
- 6) It is mandatory to include a 'Data Availability' section after the Materials and Methods. Before submitting your revision, primary

datasets produced in this study need to be deposited in an appropriate public database, and the accession numbers and database listed under 'Data Availability'. Please remember to provide a reviewer password if the datasets are not yet public (see <https://www.embopress.org/page/journal/17574684/authorguide#dataavailability>).

12) Author contributions: You will be asked to provide CRediT (Contributor Role Taxonomy) terms in the submission system. These replace a narrative author contribution section in the manuscript.

13) A Conflict of Interest statement should be provided in the main text.

14) Every published paper now includes a 'Synopsis' to further enhance discoverability. Synopses are displayed on the journal webpage and are freely accessible to all readers. They include a short stand first (maximum of 300 characters, including space) as well as 2-5 one-sentences bullet points that summarizes the paper. Please write the bullet points to summarize the key NEW findings. They should be designed to be complementary to the abstract - i.e. not repeat the same text. We encourage inclusion of key acronyms and quantitative information (maximum of 30 words / bullet point). Please use the passive voice. Please attach

these in a separate file or send them by email, we will incorporate them accordingly.

15) Include a Reagents and Tools Table as part of the Methods section, which can be downloaded from our author guidelines (<https://www.embopress.org/page/journal/17574684/authorguide#structuredmethods>)

***** Reviewer's comments *****

Referee #1 (Remarks for Author):

LCMV represents the best-investigated member of the arenavirus family. Arenaviruses including LCMV infect rodents and occasionally humans. Several arenaviruses are known to cause human disease. The diseases caused by arenaviruses range in severity, among which Lassa fever is of particular relevance due to its endemic occurrence in west Africa. The lack of a licensed vaccine and the limited number of therapeutic options available to date (only ribavirin) mean that arenaviruses are among the most neglected virus groups.

In their present study, Gorzkiewicz et al. systematically investigated the inhibitory potential of heparin and other sulphated glycosaminoglycans (GAG) on the basis of previous work published by various research groups. The authors identified dextran sulphate (DS) as the most effective inhibitor of LCMV infection in vitro. Based on these findings, the authors conducted further comprehensive analyses concerning:

1. the ultrastructure of the interaction between DS, the LCMV prefusion GP and one of the LCMV receptors, i.e. α -DG;
2. the extent of its efficacy against other arenaviruses in vitro; and
3. the impact of DS treatment on LCMV replication and immunopathology in vivo.

Due to the scope of this study, it was not possible to investigate all aspects in absolute depth. This was particularly the case with regard to the treatment studies in mice, as these carry a risk of increasing the immunopathological consequences of GAG-based intervention. Overall, this study's results are of particular technical quality. The data are novel and of broad interest, as the sulphated glycosaminoglycan (GAG) substance group is already in use in patients as a plasma expander and is able to inhibit a large number of viruses. Thus, it has therapeutic potential beyond arenaviruses, but it also carries potential risks in situations where the course of infection is governed by immunopathology.

However, the paper has also a major flaw:

The title is not supported by the experimental evidence provided. First, the claim that sulphated GAGs inhibit arenaviruses is an overstatement given the data shown in fig. 5. MORV, TCRV and LASV are very poorly if at all inhibited even by very high doses of DS and heparin. This is of particular clinical relevance since Lassa virus causes viral hemorrhagic fever with a considerably high mortality. The virus replication data on which figure 5 is based should at least be shown in the appendix of the publication. The presentation of normalised values is by no means sufficient. Second, the data document that DS inhibits virus particle attachment rather than the subsequent step of entry. Elaborate EM-based studies as shown in figure 4 are certainly valuable. However, FACS-based approaches could provide a simpler way of obtaining quantitative and kinetic data on the interaction between virus particles and target receptors on cells. The fact that LCMV utilises several cell receptors in addition to α -DG, and that LCMV isolates utilise these receptors differently, complicates the interpretation and generalisation of the data and the further exploration of DS-based treatment in vivo. Elegant tools and systems developed in previous studies could be used in the future to differentiate between virus attachment and virus entry (Volland et al., PLoS Pathogens 2021).

Further points:

Page 6, line 31: Since Tyr 155 is associated with ... (Xu et al., 2024). At this point, the citation is not correct, as previous work by other investigators has already shown this and should therefore be cited (e.g. (Volland et al., PLoS Pathogens 2021)).

Page 8, line 9: should read ... well-known markers of liver dysfunction ...

Page 14, #4.5.: information regarding the sex and age of the mice used in the experiments is not provided.

Fig. 2 D & E: change colors of circles indicating dextran vs. DS, which are hardly distinguishable

Fig. 3 E: Man70 and Man70-sulf are assigned incorrectly

Fig. 3 F and G: the data indicating the number of infected cells at high concentrations (500 ug/ml) appear not consistent, since 2.5 hrs p.i. the number of infected cells is much higher than after 24 hrs p.i.

Referee #2 (Remarks for Author):

Gorzkiwicz et al. demonstrate that sulfated glycosaminoglycans (GAGs) inhibit the infectivity of LCMV and other arenaviruses. In vivo, treatment of mice with dextran sulfate (DS) at the beginning of LCMV infection resulted in reduced early viral loads in various organs. However, this early intervention also led to a diminished T cell response, prolonged infection, and liver pathology. In contrast, DS administration during the acute phase of infection (7 days post-infection) enhanced antiviral CD8+ T cell responses in the blood and contributed to viral clearance.

Given the limited treatment options for arenavirus infections, this study represents a valuable contribution toward the development of novel therapeutic strategies against LCMV and arenaviruses more broadly.

Arenaviruses are known to use proteoglycans to attach to the host cells. Thus, inhibiting these interactions with highly sulfated GAGs represents a promising therapeutic approach. Indeed, DS inhibited LCMV infection in MC57G cells in vitro. However, the authors extrapolate their findings to arenaviruses more broadly, which should be interpreted with caution. While the abstract claims that LASV infection is inhibited by DS, the data presented in Figure 5 show clear inhibition only for LUJV, JUNV, MACV, and to a lesser extent PARV. In contrast, LASV, MORV, and TCRV infections were minimally affected, suggesting that a generalized statement regarding arenavirus inhibition by DS is not supported by the data and should be avoided throughout the manuscript. Although proteoglycan interactions are clearly important for arenavirus entry, different arenaviruses can engage distinct or additional cellular receptors, e.g. α -dystroglycan, which could account for the differential sensitivity to DS.

The authors compare the capacity of DS to inhibit LCMV infection of MC57G and BHK cells between low-affinity (rWT) and high-affinity (H115Y) variants (Figure 6 and S3). Although the authors suggest greater inhibition of the low-affinity variant, direct comparison is difficult due to differing maximal infection rates between the variants, especially in Supplementary Figure S3. Calculating and presenting IC50 values would provide a more accurate and quantitative comparison of inhibitory effects.

A significant limitation of the study is the reliance on MC57G and BHK cell lines, which are not representative of physiologically relevant target cells such as dendritic cells (DCs) or macrophages - particularly critical for studying LCMV pathogenesis. Thus, experiments with e.g. bone marrow-derived DCs, isolated spleen DCs would improve the impact of the manuscript, especially if using the low-affinity (rWT) and high-affinity (H115Y) variants, which differ in their affinity for α -dystroglycan and, likely, in their ability to infect and modulate DC function.

In Figure 1, both preincubation of cells and virus with heparin reduced infection to a similar extent. However, since unbound heparin was not removed after preincubation, it is difficult to draw definitive conclusions about the exact mechanism of inhibition. Additional controls or removal steps would strengthen conclusions regarding the mechanism of inhibition.

The authors should include viability data for the cell lines treated with the compounds, provided as supplementary material, to rule out cytotoxic effects.

Regarding Figure 5, statistical significance should be indicated. The label "viral growth (%)" is potentially misleading. The label, Virus titer relative to the solvent control would provide a clearer context.

In Figure 7C, DS appears to alter the lymphoid tissue structure, as seen in the more diffuse localization of CD169+ marginal zone macrophages (MZMs). It is unclear how representative this image is, and whether similar changes occur in the absence of infection. Since CD169+ MZMs are likely involved in LCMV-specific immune responses, further investigation into the effects of systemic DS treatment on lymphoid architecture is essential. If DS impacts host immune responses, interpreting infection outcomes becomes complex - particularly in the case of LCMV, where virus-host immune interactions are highly dynamic and central to pathogenesis.

Generally, for all flow cytometry analyses, representative gating and plots should be provided as supplementary figures to support the presented quantifications.

Referee #3 (Remarks for Author):

In the presented manuscript, Gorzkiwicz et al. investigate the effect of sulfated glycosaminoglycans on the cell entry of different types of Arenaviruses, including Lymphocytic Choriomeningitis Virus (LCMV WE). Authors report that pretreatment of cells or viral particles with glycosaminoglycan (e.g., heparin, dextran sulfate) decreases the frequency of successfully infected cells in a dose-dependent manner. They measure the sulfation levels of different glycosaminoglycans (GAGs) and utilize sulfated and non-sulfated GAGs to demonstrate sulfation-level-dependent potential of GAGs to inhibit virus entry into cells. Choosing the GAG with the highest potential for virus entry inhibition (dextran sulfate) and its non-sulfated forms, the authors continue to

investigate the inhibitory effect of sulfated GAG on the cell entry of several different pathogenic and non-pathogenic Old and New World Arenaviruses. The authors perform ligand diffusion molecular dynamics simulations to investigate the potential binding site of dextran sulfate to the virus receptors (an atomistic model of the fully glycosylated pre-fusion LCMV GP trimer). The paragraph committed to this analysis is left without a conclusion. Finally, the authors test the in vivo anti-viral effect of dextran sulfate application, using the LCMV WE infection model in mice. They perform two tests designed to estimate the effect of dextran sulfate application during 1) the early phase and 2) the acute phase of infection. The authors profile virus-specific CD8 T cells, viral titers, and serum markers of liver damage to estimate the effect of dextran sulfate on the outcome of viral infection.

The study contains numerous issues (see major comments below), which significantly diminish its novelty and potential impact on the field.

In general, sulfated glycosaminoglycans are known inhibitors of viral entry into the cells; their potential has even been tested in models of arenavirus infection (Andrei and Clercq, 1990), making the novelty of the study very limited. The manuscript is poorly written, both in terms of structure and language. The introduction must be more concise. The number of main figures is too high, with figures (e. g. 4 and 5) containing only a few items. Selecting only the essential data to be presented in the main figures, thereby reducing the number of main figures, would improve the structure and condense the message, hence improving the delivery. The discussion section is too general, poorly structured, and lacks the necessary placement of the study findings in the broader context of contemporary research in the field.

Major concerns:

1. It is not clear why the authors chose to study sulfated glycosaminoglycans specifically. Better reasoning in the introduction section would help the reader understand why it is essential to investigate the effect of exactly sulfated glycosaminoglycans on arenavirus cell entry.
2. The title of the study refers to the inhibition of Arenavirus entry by sulfated glycosaminoglycans; however, the effect of the compounds on the virus replication or other steps of the virus infection cycle has not been investigated. Despite this, upon in vivo dextran sulfate testing, the authors conclude that compound application influences viral replication. It is unclear which stage of the virus infection cycle is explicitly affected by glycosaminoglycan treatment. Is it only the virus attachment or also viral replication, virus dissociation from the receptor, virus budding, and cell-to-cell transfer? Are any of the listed stages explicitly affected by the sulfation levels of the GAGs?
3. The authors mainly use cell lines to test the in vitro effects of GAGs treatment on virus cell entry. How is glycosaminoglycan treatment affecting the infectability, antigen-presenting function, and maturation status of primary mouse dendritic cells (DCs)?
4. The experimental design of the in vivo experiments is confusing. Why is the timepoint of the early-treatment analysis (day 12 after infection - all organs and blood), different from that selected for the acute-infection-treatment (day 8)? The justification for the chosen time points is missing.
5. The conclusion to the experiments described under paragraph 2.4 is: " these data show that the treatment with dextran sulfate of LCMV particles and mice at the beginning of infection caused reduced viral replication resulting in impaired immune activation including reduced T cell function, leading to virus persistence and pathology." This conclusion is very general, what do the authors mean by "immune activation", which particular element is referred to here? Is it impaired activation of DCs, inability to present the antigens to T cells, or/and co-stimulate T cells? Is the T cell priming phase affected?
6. Does dextran sulfate treatment affect T cell recruitment to the priming site? Are T cells primed sufficiently? Are T cells not getting sufficiently activated, or do they have a delayed peak of activation? What is the status of the early T cell activation markers expression, e.g., CD44, CD69, CD62L (report as frequency of tet+ cells and MFI).
7. Is there a direct effect of dextran sulfate treatment on (antiviral) T cell function? If so, is it different during the priming and effector phases?
8. Are animals receiving the early dextran sulfate treatment able to clear the virus? What implications do these findings have for the potential treatment options?
9. Does dextran sulfate treatment affect the outcome of infection using LCMV variants with high affinity binding to the receptor (e.g., LCMV C13)?

Minor concerns:

1. Avoid using repetitive bridging words in consecutive sentences, e.g., "however" in the last two sentences of the first paragraph in the introduction section.
2. Use bigger symbols for the bar graph panels in the figures.
3. Fig 8 Legend:
B. Tet-gp33+ and tet-np396+ T cells were determined in the blood, spleen and liver.
C. SLECs (IL7R-, KLRG1+) and MPECs (IL7R+, KLRG1-) subsets within tet+ T cells were determined in the blood, spleen and liver.
(Name the value presented e.g. Number of ...)

Point-by-point list of the revisions:

We would like to thank the Reviewers for their valuable input. In the response we provide new experimental data strengthening the conclusions of the manuscript.

Referee #1 (Remarks for Author)

The title is not supported by the experimental evidence provided. First, the claim that sulphated GAGs inhibit arenaviruses is an overstatement given the data shown in fig. 5. MORV, TCRV and LASV are very poorly if at all inhibited even by very high doses of DS and heparin. This is of particular clinical relevance since Lassa virus causes viral hemorrhagic fever with a considerably high mortality. The virus replication data on which figure 5 is based should at least be shown in the appendix of the publication. The presentation of normalised values is by no means sufficient.

Answer: We would like to thank the Reviewer for this thoughtful comment. We fully agree that the original presentation of normalized values in Figure 5 did not provide sufficient details to support our conclusions. In response, we have now revised the Figure and included the underlying virus replication data (new Fig. 2I and Fig. EVIG), which allow for a direct assessment of absolute replication levels in the presence of different GAGs. This modification makes it clearer that while inhibition is modest for some arenaviruses, others (such as LUJV and JUNV) display significant sensitivity to sulfated GAGs. We have adjusted the wording in the Results and Discussion sections to better reflect these virus-specific differences and to avoid overgeneralization of the inhibitory effect across all arenaviruses.

Additionally, we provide new experimental data of viral transcripts (for LASV, LUJV and MORV) in supernatants collected in described experiments. As seen in the table below, in case of heparin and dextran sulfate for most concentrations tested, the viral transcripts are below the detection limit, while the treatment with dextran does not influence the viral load. These data are consistent with the results of replication assay presented in new Fig. 2I and Fig. EVIG.

		Heparin	Dextran	Dextran sulfate
	concentration ($\mu\text{g/ml}$)	Ct\pmSD	Ct\pmSD	Ct\pmSD
LASV	5000	undetermined	24.76 \pm 0.39	undetermined
	50	undetermined	23.23 \pm 0.47	undetermined
	0.5	23.85 \pm 0.55	23.61 \pm 0.78	24.16 \pm 0.21
LUJV	5000	undetermined	26.98 \pm 0.55	undetermined
	50	undetermined	24.33 \pm 0.74	36.87 \pm 0.79
	0.5	26.37 \pm 0.35	23.84 \pm 0.45	26.61 \pm 0.22
MORV	5000	undetermined	22.88 \pm 0.41	undetermined
	50	undetermined	21.39 \pm 1.25	undetermined
	0.5	26.76 \pm 1.92	20.98 \pm 0.61	undetermined

Second, the data document that DS inhibits virus particle attachment rather than the subsequent step of entry. Elaborate EM-based studies as shown in figure 4 are certainly valuable. However, FACS-based approaches could provide a simpler way of obtaining quantitative and kinetic data on the interaction between virus particles and target receptors on cells. The fact that LCMV

utilises several cell receptors in addition to α -DG, and that LCMV isolates utilise these receptors differently, complicates the interpretation and generalisation of the data and the further exploration of DS-based treatment in vivo. Elegant tools and systems developed in previous studies could be used in the future to differentiate between virus attachment and virus entry (Volland et al., PLoS Pathogens 2021).

*Answer: This is a very important point. To further address the topic of viral entry inhibition, we performed an additional set of experiments, in which we pretreated MC57G cells with dextran and dextran sulfate, and subsequently treated with recombinant LCMV GP-Fc (Recher et al. 2004, Khairnar et al. 2015). Following washing steps and additional staining with PE-conjugated goat anti-human anti-Fc antibody, we were able to visualise and quantify the LCMV-GP binding to the cell surface. While dextran does not inhibit LCMV GP binding to the cell, dextran sulfate significantly decreases this binding. These new data pointing to inhibition of binding as antiviral mechanism of GAGs have been incorporated in the **Fig. 3F+G**. Additionally, we performed an entry assay of LCMV WE and clone 13 (the latter with high alpha-dystroglycan affinity) in WT and alpha-dystroglycan deficient HEK293T cells, and in Vero cells (known for the lack of proper (functional) alpha-dystroglycan glycosylation). These data indicate that regardless of cell type and virus entry rate (depending on the type and availability of cell surface receptor), dextran sulfate was able to inhibit the viral cell entry in all cases. Once again, dextran sulfate was more efficient towards low affinity virus, in line with the effect we observed for rWT and H155Y variants. These new experimental data were incorporated in the **Appendix Fig. S3E-J**.*

Page 6, line 31: Since Tyr 155 is associated with ... (Xu et al., 2024). At this point, the citation is not correct, as previous work by other investigators has already shown this and should therefore be cited (e.g. (Volland et al., PLoS Pathogens 2021)).

Answer: We would like to thank the Reviewers for this remark. We added this reference to the text.

Page 8, line 9: should read ... well-known markers of liver dysfunction ...

Answer: The text has been corrected.

Page 14, #4.5.: information regarding the sex and age of the mice used in the experiments is not provided.

Answer: We would like to thank the Reviewers for this remark. The information has been added to Materials and Methods section.

Fig. 2 D&E: change colors of circles indicating dextran vs. DS, which are hardly distinguishable

*Answer: The colors have been changed, and the data are now incorporated in the **Fig. EV1B-E**, showing only one concentration tested, in a manner similar to **Fig. 1D-E**, in order to make our message more clear.*

Fig. 3 E: Man70 and Man70-sulf are assigned incorrectly.

Answer: We would like to thank the Reviewers for this remark. Indeed, there have been a mistake, which now was corrected.

Fig. 3 F and G: the data indicating the number of infected cells at high concentrations (500 ug/ml) appear not consistent, since 2.5 hrs p.i. the number of infected cells is much higher than after 24 hrs p.i.

Answer: We would like to thank the Reviewer for pointing out these inaccuracies, which have now been revised and corrected. Regarding Fig. 3F and G, these data have been reanalyzed and presented in corrected manner in revised Fig. 2H and Fig. EVIF.

Referee #2 (Remarks for Author):

Arenaviruses are known to use proteoglycans to attach to the host cells. Thus, inhibiting these interactions with highly sulfated GAGs represents a promising therapeutic approach. Indeed, DS inhibited LCMV infection in MC57G cells in vitro. However, the authors extrapolate their findings to arenaviruses more broadly, which should be interpreted with caution. While the abstract claims that LASV infection is inhibited by DS, the data presented in Figure 5 show clear inhibition only for LUJV, JUNV, MACV, and to a lesser extent PARV. In contrast, LASV, MORV, and TCRV infections were minimally affected, suggesting that a generalized statement regarding arenavirus inhibition by DS is not supported by the data and should be avoided throughout the manuscript.

Answer: We would like to thank the Reviewer for this thoughtful comment. We fully agree that the original presentation of normalized values in Figure 5 did not provide sufficient detail to support our conclusions. In response, we have now revised the Figure and included the underlying virus replication data (new Fig. 2I and Fig. EVIG), which allows for a direct assessment of absolute replication levels in the presence of different GAGs. This modification makes it clearer that while inhibition is modest for some arenaviruses, others (such as LUJV and JUNV) display significant sensitivity to sulfated GAGs. We have adjusted the wording in the Results and Discussion sections to better reflect these virus-specific differences and to avoid overgeneralization of the inhibitory effect across all arenaviruses.

Additionally, we provide new experimental data of viral transcripts (for LASV, LUJV and MORV) in supernatants collected in described experiments. As seen in the table below, in case of heparin and dextran sulfate for most concentrations tested, the viral transcripts are below the detection limit, while the treatment with dextran does not influence the viral load. These data are consistent with the results of replication assay presented in new Fig. 2I and Fig. EVIG.

		Heparin	Dextran	Dextran sulfate
	concentration (µg/ml)	Ct±SD	Ct±SD	Ct±SD
LASV	5000	undetermined	24.76±0.39	undetermined
	50	undetermined	23.23±0.47	undetermined
	0.5	23.85±0.55	23.61±0.78	24.16±0.21
LUJV	5000	undetermined	26.98±0.55	undetermined
	50	undetermined	24.33±0.74	36.87±0.79
	0.5	26.37±0.35	23.84±0.45	26.61±0.22
MORV	5000	undetermined	22.88±0.41	undetermined
	50	undetermined	21.39±1.25	undetermined
	0.5	26.76±1.92	20.98±0.61	undetermined

Although proteoglycan interactions are clearly important for arenavirus entry, different arenaviruses can engage distinct or additional cellular receptors, e.g. α -dystroglycan, which could account for the differential sensitivity to DS.

*Answer: To further address this issue, we performed an entry assay of LCMV WE and clone 13 (the latter with high α -dystroglycan affinity) in WT and α -dystroglycan deficient HEK293T cells, and in Vero cells (known for the lack of proper (functional) α -dystroglycan glycosylation). These data indicate that regardless of cell type and virus entry rate (depending on the type and availability of cell surface receptor), dextran sulfate was able to inhibit the viral cell entry in all cases. Once again, dextran sulfate was more efficient towards low affinity virus (LCMV WE), in line with the effect we observed for rWT and H155Y variants. The new experimental data are incorporated in the **Appendix Fig. S3E-J**.*

The authors compare the capacity of DS to inhibit LCMV infection of MC57G and BHK cells between low-affinity (rWT) and high-affinity (H155Y) variants (Figure 6 and S3). Although the authors suggest greater inhibition of the low-affinity variant, direct comparison is difficult due to differing maximal infection rates between the variants, especially in Supplementary Figure S3. Calculating and presenting IC50 values would provide a more accurate and quantitative comparison of inhibitory effects.

*Answer: This is a very important point. IC50 values have been calculated and added to **Fig. 4A-B, Fig. 6A, Appendix Fig. S3**.*

A significant limitation of the study is the reliance on MC57G and BHK cell lines, which are not representative of physiologically relevant target cells such as dendritic cells (DCs) or macrophages - particularly critical for studying LCMV pathogenesis. Thus, experiments with e.g. bone marrow-derived DCs, isolated spleen DCs would improve the impact of the manuscript, especially if using the low-affinity (rWT) and high-affinity (H155Y) variants, which differ in their affinity for α -dystroglycan and, likely, in their ability to infect and modulate DC function.

*Answer: We would like to thank the Reviewer for this suggestion. In new experiments we performed entry assays in BMDCs using rWT and H155Y viruses (**Fig. 6A**). According to the new experimental data, dextran sulfate inhibited cell entry of both viruses in a similar manner observed previously: dextran sulfate had more potent inhibitory activity towards the low-affinity LCMV variant. Moreover, the high-affinity variant showed slightly higher infectivity of BMDCs, consistent with previous reports (Sevilla et al. 2003). These data are incorporated into the manuscript.*

In Figure 1, both preincubation of cells and virus with heparin reduced infection to a similar extent. However, since unbound heparin was not removed after preincubation, it is difficult to draw definitive conclusions about the exact mechanism of inhibition. Additional controls or removal steps would strengthen conclusions regarding the mechanism of inhibition.

*Answer: We would like to thank the Reviewer for this thoughtful comment, this is indeed a very important point. We performed additional experiments pretreating LCMV WE with dextran sulfate, followed by removal of excess unbound GAG by filtration. Subsequent entry assays in MC57G cells showed that filtration itself does not directly influence the infectivity of LCMV, while the majority of dextran sulfate remains bound to the virus, providing inhibition of infection (**Fig. 3E**). These data are incorporated into the manuscript.*

The authors should include viability data for the cell lines treated with the compounds, provided as supplementary material, to rule out cytotoxic effects.

Answer: This is an important point. Additional analyses (resazurin-based cytotoxicity assay and Annexin V/7AAD staining) have been performed (Appendix Fig. S2). We were able to show that studied GAGs are non-toxic to the cells of studied cell lines, excluding the direct impact of cytotoxicity on viral entry and replication.

Regarding Figure 5, statistical significance should be indicated. The label "viral growth (%)" is potentially misleading. The label, Virus titer relative to the solvent control would provide a clearer context.

Answer: We have now changed the data presentation (as indicated above), and added statistical analyses.

In Figure 7C, DS appears to alter the lymphoid tissue structure, as seen in the more diffuse localization of CD169+ marginal zone macrophages (MZMs). It is unclear how representative this image is, and whether similar changes occur in the absence of infection. Since CD169+ MZMs are likely involved in LCMV-specific immune responses, further investigation into the effects of systemic DS treatment on lymphoid architecture is essential. If DS impacts host immune responses, interpreting infection outcomes becomes complex - particularly in the case of LCMV, where virus-host immune interactions are highly dynamic and central to pathogenesis.

Answer: We would like to thank the Reviewer for pointing out this important issue. We reexamined our histology analysis and have now shown more representative pictures (now in Fig. 5C). We also quantified the overall CD169 expression in these samples (Fig. 5F), showing that it remains unchanged after infection in the presence of dextran sulfate. Therefore, at this stage we conclude that DS dose not directly influence the lymphoid architecture. Here, we show additional representative pictures below for better overview.

Generally, for all flow cytometry analyses, representative gating and plots should be provided as supplementary figures to support the presented quantifications.

*Answer: We have now included the representative gating strategies and plots in **Appendix Fig. S7**.*

Referee #3 (Remarks for Author):

In the presented manuscript, Gorzkiewicz et al. investigate the effect of sulfated glycosaminoglycans on the cell entry of different types of Arenaviruses, including Lymphocytic Choriomeningitis Virus (LCMV WE). Authors report that pretreatment of cells or viral particles with glycosaminoglycan (e.g., heparin, dextran sulfate) decreases the frequency of successfully infected cells in a dose-dependent manner. They measure the sulfation levels of different glycosaminoglycans (GAGs) and utilize sulfated and non-sulfated GAGs to demonstrate sulfation-level-dependent potential of GAGs to inhibit virus entry into cells. Choosing the GAG with the highest potential for virus entry inhibition (dextran sulfate) and its non-sulfated forms, the authors continue to investigate the inhibitory effect of sulfated GAG on the cell entry of several different pathogenic and non-pathogenic Old and New World Arenaviruses.

The authors perform ligand diffusion molecular dynamics simulations to investigate the potential binding site of dextran sulfate to the virus receptors (an atomistic model of the fully glycosylated pre-fusion LCMV GP trimer). The paragraph committed to this analysis is left without a conclusion.

Answer: We would like to thank the Reviewer for pointing out this important issue. We have now added the conclusion to this part of the manuscript.

Finally, the authors test the in vivo anti-viral effect of dextran sulfate application, using the LCMV WE infection model in mice. They perform two tests designed to estimate the effect of dextran sulfate application during 1) the early phase and 2) the acute phase of infection. The authors profile virus-specific CD8 T cells, viral titers, and serum markers of liver damage to estimate the effect of dextran sulfate on the outcome of viral infection.

The study contains numerous issues (see major comments below), which significantly diminish its novelty and potential impact on the field.

In general, sulfated glycosaminoglycans are known inhibitors of viral entry into the cells; their potential has even been tested in models of arenavirus infection (Andrei and Clercq, 1990), making the novelty of the study very limited. The manuscript is poorly written, both in terms of structure and language. The introduction must be more concise.

Answer: We have now revised and rewritten significant portions of the manuscript, as the Reviewer raised several critical points. The rewritten Introduction is shorter, and both the Introduction and Discussion flow more logically and directly address the rationale and topics of our studies, acknowledging previous reports and emphasizing our findings.

*We also performed additional analysis (filtration assay and GP-Fc binding assay (**Fig. 3**), impact of dextran sulfate treatment on the infectivity of BMDCs and subsequent activation of T cells (**Fig. 6, Fig. EV2**), and the role of dextran sulfate treatment during LCMV clone 13*

infection (Fig. EV5). We believe that these new data provide additional insight and strengthen the manuscript.

The number of main figures is too high, with figures (e. g. 4 and 5) containing only a few items. Selecting only the essential data to be presented in the main figures, thereby reducing the number of main figures, would improve the structure and condense the message, hence improving the delivery. The discussion section is too general, poorly structured, and lacks the necessary placement of the study findings in the broader context of contemporary research in the field.

Answer: We would like to thank the Reviewer for this suggestion. We have now restructured the manuscript and provide a more concise revised version of the manuscript. We have also rewritten and restructured the Discussion to highlight the goal and impact of our studies.

Major concerns:

1. It is not clear why the authors chose to study sulfated glycosaminoglycans specifically. Better reasoning in the introduction section would help the reader understand why it is essential to investigate the effect of exactly sulfated glycosaminoglycans on arenavirus cell entry.

Answer: We would like to thank the Reviewer for this suggestion. We have now included the explanation in the Introduction, which we believe indicates more closely the importance of studying GAGs as antivirals.

2. The title of the study refers to the inhibition of Arenavirus entry by sulfated glycosaminoglycans; however, the effect of the compounds on the virus replication or other steps of the virus infection cycle has not been investigated. Despite this, upon in vivo dextran sulfate testing, the authors conclude that compound application influences viral replication. It is unclear which stage of the virus infection cycle is explicitly affected by glycosaminoglycan treatment. Is it only the virus attachment or also viral replication, virus dissociation from the receptor, virus budding, and cell-to-cell transfer? Are any of the listed stages explicitly affected by the sulfation levels of the GAGs?

Answer: We thank the Reviewer for this important remark. We fully agree that our study does not directly address all stages of the arenavirus replication cycle, particularly those occurring beyond the initial entry phase. Our data are most consistent with an effect at the earliest step, namely virus binding and entry. To strengthen this conclusion, we have now included additional experimental analyses – specifically a filtration assay and a GP-Fc binding assay (Fig. 3F+G) – which demonstrate that sulfated GAGs can bind to arenaviral particles and thereby interfere with their attachment to host cells. We therefore conclude that the principal impact of sulfated GAGs in our system lies at the entry step, while effects on later stages of infection remain beyond the scope of this study.

3. The authors mainly use cell lines to test the in vitro effects of GAGs treatment on virus cell entry. How is glycosaminoglycan treatment affecting the infectability, antigen-presenting function, and maturation status of primary mouse dendritic cells (DCs)?

Answer: This is indeed a valid point. We have now evaluated the impact of dextran sulfate treatment on the infectivity of BMDCs, together with subsequent activation of CD8⁺ P14⁺ T cells (Fig. 6, Fig. EV2). We conclude that dextran sulfate administered at the early stages of infection can block the infection of dendritic cells, which translates to decreased antigen

presentation and activation of T cells. These new experimental data are incorporated into the manuscript.

4. The experimental design of the in vivo experiments is confusing. Why is the timepoint of the early-treatment analysis (day 12 after infection - all organs and blood), different from that selected for the acute-infection-treatment (day 8)? The justification for the chosen time points is missing.

Answer: This is an important point. We have now rephrased the text providing the rationale for this experimental setup. In the case of this experimental setting, 8 days p.i. represents the peak phase of active infection, while by day 12 p.i. the virus is already partially controlled. Therefore, for 'acute-infection-treatment' (with application of dextran sulfate on day 6 and 7 p.i.) experiment we performed the analyses on day 8 p.i. to evaluate an immediate effect of dextran sulfate treatment. On the other hand, for 'early-treatment analysis', we analyzed T cell populations and titer in blood on day 8 p.i., and all the documented parameters on day 12 p.i., when according to our expectations, the virus was already partially cleared in the control group.

5. The conclusion to the experiments described under paragraph 2.4 is: " these data show that the treatment with dextran sulfate of LCMV particles and mice at the beginning of infection caused reduced viral replication resulting in impaired immune activation including reduced T cell function, leading to virus persistence and pathology." This conclusion is very general, what do the authors mean by "immune activation", which particular element is referred to here? Is it impaired activation of DCs, inability to present the antigens to T cells, or/and co-stimulate T cells? Is the T cell priming phase affected?

Answer: This is an important point. We now provide new experimental data showing that dextran sulfate treatment can block the virus entry into bone marrow-derived dendritic cells, causing their decreased activation (Fig. 6, Fig. EV2) and inability to activate CD8⁺ P14⁺ T cells. These data are incorporated into the manuscript.

6. Does dextran sulfate treatment affect T cell recruitment to the priming site? Are T cells primed sufficiently? Are T cells not getting sufficiently activated, or do they have a delayed peak of activation? What is the status of the early T cell activation markers expression, e.g., CD44, CD69, CD62L (report as frequency of tet⁺ cells and MFI).

Answer: This is an important point. Our additional analyses suggest that during early administration, dextran sulfate may inhibit LCMV infection in antigen presenting cells, therefore T cells may be not sufficiently primed. Analyzing the expression of CD44 and CD62L in blood (day 8 and 12 p.i.), spleen and liver (day 12 p.i.), we can see that especially on day 12 p.i. the expression of CD62L remains significantly higher in the dextran sulfate-treated group, suggesting decreased amount of effector T cells. On the other hand, high expression of both CD62L and PD1 (which was also observed in this experiment for dextran sulfate-treated group, Fig. EV3) is a hallmark of precursors of exhausted T cells ('T_{PEX} cells'). Since also CD44 expression is higher in this group, we may conclude that these cells have been activated by antigen and are antigen-experienced. Combined with PD-1^{high} and CD62L^{high}, it points toward a stem-like, precursor exhausted subset. This observation is quite interesting since such populations are observed during chronic LCMV infections, e.g. LCMV clone 13, and points to chronic antigen stimulation of T cells, which is understandable considering that in dextran sulfate-treated group, LCMV WE is still present in organs on day 12 p.i.

Regarding the administration of dextran sulfate during acute phase of LCMV infection, we did not observe changes of CD44 and CD62L expression on day 8 p.i., which may be attributed to quite short time after dextran sulfate treatment (1-2 days).

The expression of CD44, CD62L, and also IL7R and KLRG1 (used for determination of SLEC and MPEC subsets, reported in main figures) has been reported as frequency of tet⁺ cells and MFI, and included in **Appendix Fig. S8 and S9**.

7. Is there a direct effect of dextran sulfate treatment on (antiviral) T cell function? If so, is it different during the priming and effector phases?

*Answer: In order supplement our experiment with the use of BDMCs and CD8⁺ P14⁺ T cells, where we observed impaired T cell activation after treatment of BMDCs with dextran sulfate before LCMV infection, we isolated CD8⁺ T cells from C57BL/6J mice, activated them with anti-CD3/CD28 antibodies, and at the same time treated with dextran or dextran sulfate. For proliferation analysis, the cells were additionally stained with cell proliferation dye. After 48h, the cells were stained with anti-CD8, -CD62L, -CD44 antibodies and analyzed for proliferation and T cells populations (**Fig. EV2D+E**). While we observed slight but significant increase in T cell proliferation and differentiation, suggesting a rather direct positive impact of dextran sulfate on T cells – we did not observe any impairment in T cell proliferation in this setting.*

8. Are animals receiving the early dextran sulfate treatment able to clear the virus? What implications do these findings have for the potential treatment options?

Answer: To answer this question, we performed new experiments infecting C57BL/6J mice with LCMV WE pretreated with dextran sulfate. Mice were re-injected i.v. with dextran sulfate 6h p.i.; at the same time, while the control group was injected with PBS following infection of LCMV WE. Analyses were carried out on day 20 p.i..

We observed that animals receiving early dextran sulfate treatment are able to clear the virus, although clearance occurs with some delay compared to untreated controls: in spleen and liver the virus was completely cleared, with residual titers still detectable for some animals in lung, kidney, brain and blood (see below). Since these titers were analyzed on day 20 p.i., we cannot say with certainty whether the virus will be completely removed after a longer period of time. This suggests that dextran sulfate does not completely prevent the development of effective antiviral immunity, but may alter the kinetics of viral control. From a therapeutic perspective, considering that dextran sulfate administered during the early phase of infection might cause prolonged viral infection, these findings indicate that dextran sulfate application would need to be carefully timed to ensure both effective suppression of infection and timely virus clearance.

*LCMV titers were determined in organs and blood collected on day 20 p.i.. Data presented as mean±SEM, n=4 mice per condition, *p<0.0. Statistical significance was assessed by Student's t-test.*

9. Does dextran sulfate treatment affect the outcome of infection using LCMV variants with high affinity binding to the receptor (e.g., LCMV C13)?

Answer: To answer this question, C57BL/6J mice were infected with LCMV clone 13 (10^6 pfu per mouse). On day 6 and 7 p.i., mice were injected i.v. with dextran sulfate (100 μ g per mouse); at the same time, control group was injected with PBS. Analyses were carried out on day 8, 12 and 20 p.i. (Fig. EV5). On day 8 p.i. we observed increased cytokine production and lowered viral titers in the blood of mice treated with dextran sulfate. This effect however did not persist in time, and on day 20 p.i. we did not observe any differences in T cells population and organ titer. Since LCMV clone 13 is a high-affinity virus, we hypothesize that repeated administration of dextran sulfate and later days p.i. could potentially result in viral clearance.

Minor concerns:

1. Avoid using repetitive bridging words in consecutive sentences, e.g., "however" in the last two sentences of the first paragraph in the introduction section.

Answer: We have now rewritten the manuscript with the aim to simplify the language and make our message clearer.

2. Use bigger symbols for the bar graph panels in the figures.

Answer: The graphs have been corrected in order to make them more readable.

3. Fig 8 Legend:

B. Tet-gp33⁺ and tet-np396⁺ T cells were determined in the blood, spleen and liver.

C. SLECs (IL7R⁻, KLRG1⁺) and MPECs (IL7R⁺, KLRG1⁻) subsets within tet⁺ T cells were determined in the blood, spleen and liver.

(Name the value presented e.g. Number of ...)

Answer: The figure legends have been revised.

References

Recher, M., Lang, K. S., Hunziker, L., Freigang, S., Eschli, B., Harris, N. L., ... Zinkernagel, R. M. (2004). Deliberate removal of T cell help improves virus-neutralizing antibody production. *Nature immunology*, 5(9), 934-942.

Khairnar, V., Duhan, V., Maney, S. K., Honke, N., Shaabani, N., Pandyra, A. A., ... Lang, K. S. (2015). CEACAM1 induces B-cell survival and is essential for protective antiviral antibody production. *Nature communications*, 6(1), 6217.

Sevilla, N., Kunz, S., McGavern, D., Oldstone, M. B. A. (2003). Infection of dendritic cells by lymphocytic choriomeningitis virus. *Dendritic Cells and Virus Infection*, 125-144.

8th Dec 2025

Dear Prof. Lang,

Thank you for the submission of your manuscript to EMBO Molecular Medicine and please accept my apologies for the delay in getting back to you, which is because one referee needed more time to complete his/her review. I am pleased to inform you that we will be able to accept your manuscript pending the following final amendments:

- 1) Please address referee #2 minor suggestions.
- 2) Figures: Please submit main Figures and EV Figures as individual, high resolution figure files. Please place figure legends for main and EV figures at the end of the main manuscript file. Correct the heading from Figure Captions to Figure Legends and add a heading for the EV Figures: "Expanded View Figure Legends". Please check "Author Guidelines" for more information:
<https://www.embopress.org/page/journal/17574684/authorguide#figureformat>
<https://www.embopress.org/page/journal/17574684/authorguide#expandedview>
- 3) Source data:
 - During our standard source data analysis, we note that numerical values in several source files for different figures are duplicated (see attached excel files). We would like to clarify these issues before we proceed with publication of your manuscript. We kindly invite you to check attached source data excel files with identified duplicated values that are color labeled (you can ignore the color scheme) and clarify the cause of these duplications.
 - Please upload source data for EV figures and Appendix figures as one zipped folder.
- 4) In the main manuscript file, please do the following:
 - Please address all comments suggested by our data editors listed below:
 - o Figure legends:
 1. Please note that the exact p values are not provided in the legends of figures 1A-E, F-J; 2D-F, H, I; 3C, E, G; 4A, B; 5B, D-G; 6A, D, F, G, H, I, J; 7B, C, D, E, H, J, K, L; EV1 B-G; EV2 B, C, E, EV3 A-C, D-I; EV4 C, D, F; EV5 I, J, K, M.
 - Limit keywords to max 5.
 - In Methods, add statistical paragraph that should reflect all information that you have filled in the Authors Checklist, especially regarding randomization, blinding, replication.
 - In Methods, add a dedicated "Graphics" section following this format (remove BioRender reference from figure legends):
Graphics:
(some of the... OR Figure #... OR synopsis) Graphics were created with BioRender.com.
 - Remove the appendix figure legends from the manuscript text.
 - Indicate in legends number and nature of replicates and exact p= values, not a range, along with the statistical test used. To keep the figures "clear" some authors found providing an Appendix table Sx with all exact p-values preferable. You are welcome to do this if you want to.
 - Correct the reference citation in the reference list. Where there are more than 10 authors on a paper, 10 will be listed, followed by "et al.". Also, please remove DOIs for published articles. Please check "Author Guidelines" for more information.
<https://www.embopress.org/page/journal/17574684/authorguide#referencesformat>
- 5) Appendix: Please merge "Appendix Figures" and "Appendix" files to one PDF file named Appendix and add table of contents on the title page. Please place the figure legends underneath the corresponding figures. The "1. Dextran sulfation - elemental analysis and NMR spectrum" and "2. Synthesis of Man70 and Man70-sulf polymers" should be placed under the heading "Appendix Supplementary Information" and appropriately cited in the main manuscript text. Please upload a "clean" file as the appendix will not be typeset and will be published as is.
- 6) Tables: Please place Table 1 after the main figure legends.
- 7) Funding: Please ensure that all relevant funders are listed in both the manuscript and in our system. Currently Jurgen Manchot Foundation (Molecules of Infection), the Volkswagen Foundation, and the Medical Faculty of the Heinrich Heine University are missing in our system. Please correct.
- 8) The Paper Explained: Please rename "Summary" to "The Paper Explained" and add it to the main manuscript file.
- 9) Synopsis:
 - Synopsis image: Please resize the image to 550 px-wide x 300-600 pixels high and upload it as a high-resolution .jpeg or .png file.
 - Please check your synopsis text and image before submission with your revised manuscript. Please be aware that in the proof stage minor corrections only are allowed (e.g., typos).
- 10) As part of the EMBO Publications transparent editorial process (see our Editorial at <http://embomolmed.embopress.org/content/2/9/329>), EMBO Molecular Medicine will publish online a Review Process File (RPF) to accompany accepted manuscripts. This file will be published in conjunction with your paper and will include the anonymous referee reports, your point-by-point response and all pertinent correspondence relating to the manuscript. Let us if you want to remove or not any figures from it prior to publication. Please note that the Authors checklist will be published at the end of the RPF.
- 11) Please provide a point-by-point letter INCLUDING my comments as well as the reviewer's reports and your detailed responses (as Word file).

I look forward to reading a new revised version of your manuscript as soon as possible.

Yours sincerely,

Zeljko Durdevic

Zeljko Durdevic
Senior Editor
EMBO Molecular Medicine

*** Instructions to submit your revised manuscript ***

When preparing your revised manuscript, please refer to our guidelines: <https://link.springer.com/journal/44321/submission-guidelines#cms-Revised-submissions>. We perform an initial quality control of all revised manuscripts before re-review; failure to include requested items will delay the evaluation of your revision.

We require:

- 1) A .docx formatted version of the manuscript text (including legends for main figures, EV figures and tables). Please make sure that the changes are highlighted to be clearly visible.
- 2) Individual production quality figure files as .eps, .tif, .jpg (one file per figure). For guidance, download the 'Figure Guide PDF': <https://media.springernature.com/original/springer-cms/rest/v1/content/27825798/data/v1>.
- 3) A .docx formatted letter INCLUDING the reviewers' reports and your detailed point-by-point responses to their comments. As part of the EMBO Press transparent editorial process, the point-by-point response is part of the Review Process File (RPF), which will be published alongside your paper.
- 4) A complete author checklist, which you can download from our author guidelines. Please insert information in the checklist that is also reflected in the manuscript. The completed author checklist will also be part of the RPF.
- 5) Please note that all corresponding authors are required to supply an ORCID ID for their name upon submission of a revised manuscript.
- 6) It is mandatory to include a 'Data Availability' section after the Materials and Methods. Before submitting your revision, primary datasets produced in this study need to be deposited in an appropriate public database, and the accession numbers and database listed under 'Data Availability'. Please remember to provide a reviewer password if the datasets are not yet public.

- 7) For data quantification: please specify the name of the statistical test used to generate error bars and P values, the number (n) of independent experiments (specify technical or biological replicates) underlying each data point and the test used to calculate p-values in each figure legend. The figure legends should contain a basic description of n, P and the test applied.

Graphs must include a description of the bars and the error bars (s.d., s.e.m.).

9) Our journal encourages inclusion of *data citations in the reference list* to directly cite datasets that were re-used and obtained from public databases. Data citations in the article text are distinct from normal bibliographical citations and should directly link to the database records from which the data can be accessed. In the main text, data citations are formatted as follows: "Data ref: Smith et al, 2001" or "Data ref: NCBI Sequence Read Archive PRJNA342805, 2017". In the Reference list, data citations must be labeled with "[DATASET]". A data reference must provide the database name, accession number/identifiers and a resolvable link to the landing page from which the data can be accessed at the end of the reference.

- the medical issue you are addressing,

- the results obtained and

- their clinical impact.

12) Author contributions: You will be asked to provide CRediT (Contributor Role Taxonomy) terms in the submission system. These replace a narrative author contribution section in the manuscript.

13) A Disclosure and competing interests statement should be provided in the main text.

14) Every published paper includes a 'Synopsis' to further enhance discoverability. Synopses are displayed on the journal webpage and are freely accessible to all readers. They include a short stand first (maximum of 300 characters, including space) as well as 2-5 one-sentences bullet points that summarizes the paper. Please write the bullet points to summarize the key NEW findings. They should be designed to be complementary to the abstract - i.e. not repeat the same text. We encourage inclusion of key acronyms and quantitative information (maximum of 30 words / bullet point). Please use the passive voice. Please attach these in a separate file or send them by email, we will incorporate them accordingly.

15) Include a Reagents and Tools Table as part of the Methods section, which can be downloaded from our author guidelines.

Photos 400-800 DPI

*Additional important information regarding figures and illustrations can be found at <https://media.springernature.com/original/springer-cms/rest/v1/content/27825798/data/v1>

***** Reviewer's comments *****

Referee #1 (Remarks for Author):

EMM-2025-21985-V2

The authors have fully addressed the comments of this reviewer, generated new data (see Fig. 2I, Fig. EV1G, Fig. 3F and Fig. 3G) and used this data to substantiate their conclusions. The errors that were criticised have been corrected, including in Figures 2H and EV1F.

Referee #2 (Comments on Novelty/Model System for Author):

GAGs have been extensively studied, however the presented manuscript reveals some novel aspects, that are of interest to the scientific community.

Referee #2 (Remarks for Author):

The authors satisfactorily addressed all of my concerns. Overall, the manuscript has been significantly improved based on reviewers' comments.

Only a few minor suggestions:

Introduction Page 4. „Sulfated glycosaminoglycans, including heparan sulfate and its analogues, are of particular interest because their unique biochemical properties, particularly their dense negative charge, enable strong interactions with viral proteins, thus preventing their binding to specific cell surface receptors." The wording 'strong' should be avoided, as electrostatic interactions are rather low-affinity.

Appendix Figure S2 E-M. Changing the color of the letters in dark blue areas to white would improve readability.

Referee #3 (Remarks for Author):

The manuscript has been significantly improved in structure and writing quality. The authors have added new experimental evidence, which improved the clarity of the study. This reviewer has no further complaints.

Point-by-point list of the revisions:**Referee #2 (Remarks for Author):**

Introduction Page 4. „Sulfated glycosaminoglycans, including heparan sulfate and its analogues, are of particular interest because their unique biochemical properties, particularly their dense negative charge, enable strong interactions with viral proteins, thus preventing their binding to specific cell surface receptors." The wording 'strong' should be avoided, as electrostatic interactions are rather low-affinity.

Answer: We would like to thank the Reviewer for this remark. The wording has been changed to avoid confusion.

Appendix Figure S2 E-M. Changing the color of the letters in dark blue areas to white would improve readability.

Answer: We would like to thank the Reviewer for this suggestion. We have adjusted the graph to the style used throughout the manuscript to improve readability.

27th Jan 2026

Dear Prof. Lang,

We are pleased to inform you that your manuscript is accepted for publication and is now being sent to our publisher to be included in the next available issue of EMBO Molecular Medicine.

You may qualify for financial assistance for your publication charges - either via a Springer Nature fully open access agreement or an EMBO initiative. Check your eligibility: <https://link.springer.com/journal/44321/how-to-publish-with-us>

Zeljko Durdevic
Senior Editor
EMBO Molecular Medicine

>>> Please note that it is EMBO Molecular Medicine policy for the transcript of the editorial process (containing referee reports and your response letter) to be published as an online supplement to each paper. If you do NOT want this, you will need to inform the Editorial Office via email immediately. More information is available here: <https://link.springer.com/partners/embo-press/editorial-policies#Peer%20review>